# An Information-Theoretic Framework for Deep Learning

**Hong Jun Jeon**
Department of Computer Science
Stanford University
Stanford, CA 94305
hjjeon@stanford.edu

**Benjamin Van Roy**
Department of Electrical Engineering
Department of Management
Science and Engineering
Stanford University
Stanford, CA 94305
bvr@stanford.edu

## Abstract

Each year, deep learning demonstrates new and improved empirical results with deeper and wider neural networks. Meanwhile, with existing theoretical frameworks, it is difficult to analyze networks deeper than two layers without resorting to counting parameters or encountering sample complexity bounds that are exponential in depth. Perhaps it may be fruitful to try to analyze modern machine learning under a different lens. In this paper, we propose a novel information-theoretic framework with its own notions of regret and sample complexity for analyzing the data requirements of machine learning. We use this framework to study the sample complexity of learning from data generated by deep ReLU neural networks and deep networks that are infinitely wide but have a bounded sum of weights. We establish that the sample complexity of learning under these data generating processes is at most linear and quadratic, respectively, in network depth.

## 1 Introduction

In modern machine learning, the apparent capabilities of empirical methods have rapidly outpaced what is soundly understood theoretically. Modern neural network architectures have scaled immensely in both parameter count and depth. GPT3, for example, encodes about 175 billion parameters in 96 decoder blocks, each with many layers within. Yet, contrary to traditional intuition, these deep neural networks with gargantuan parameter counts are able to generalize well and produce useful models with tractable amounts of data. This gap between what has been shown theoretically versus empirically makes it quite enticing to develop a coherent framework that could potentially explain this phenomenon.

Existing theory breaks down when trying to explain learning under models that are simultaneously very deep (many layers) and wide (many hidden units per layer). Classical results such as those of Haussler [1992] and Bartlett et al. [1998] can potentially handle the deep but narrow case. These results bound the sample complexity of a learning a neural network function in terms of the number of parameters and the depth. More recently, Harvey et al. [2017] established a general result that suggests that for neural networks with piecewise-linear activation units, the sample complexity grows linearly in the product of parameter count and depth.

However, when we consider neural networks with arbitrary width, these bounds become vacuous. As an alternative to parameter count methods, researchers have produced sample complexity bounds

that depend on the product of norms of realized weight matrices. Bartlett et al. [2017] and Neyshabur et al. [2018], for example, establish sample complexity bounds that scale with the product of spectral norms. Neyshabur et al. [2015] and Golowich et al. [2018] establish similar bounds that instead scale in the product of Frobenius norms. While this line of work provides sample complexity bounds that are *width*-independent, they pay for it via an exponential dependence on *depth*, which is also inconsistent with empirical results.

A large line of work has tried to ameliorate this exponential depth dependence via so-called *data-dependent* quantities Dziugaite and Roy [2017], Arora et al. [2018], Nagarajan and Kolter [2018], Wei and Ma [2019]. Among these, the most relevant to our work is Wei and Ma [2019], which bounds sample complexity as a function of depth and statistics of trained neural network weights. While difficult to interpret due to dependence on complicated data-dependent statistics, their bound suggests a nonic dependence on depth. Arora et al. [2018] also utilize concepts of compression in their analysis, which we generalize and expand upon. While they establish a sample complexity bound that suggests quadratic dependence on depth, further dependence may be hidden in data-dependent terms.

We suspect that the looseness of these results in comparison to empirical findings are due to a worst-case analysis framework. In this paper, we study an average-case notion of regret and sample complexity that is motivated by information theory. Our information-theoretic framework generalizes that developed by Haussler et al. [1994], which provided a basis for understanding the relationship between prediction error and information. In a similar vein, Russo and Zou [2019] introduced tools that establish general relationships between mutual information and error. Using these results, Xu and Raginsky [2017] established upper bounds on the generalization error of learning algorithms with countably infinite hypothesis spaces. We extend these results in several directions to enable analysis of data generating processes related to deep learning. For example, the results of Haussler et al. [1994] do not address noisy observations, and all three aforementioned papers do not accommodate continuous parameter spaces, let alone nonparametric data generating processes. A distinction of our work is that it builds on rate-distortion theory to address these limitations. While Nokleby et al. [2016] also use rate-distortion theory to study bayes risk, these results are again limited to parametric classification and only offer lower bounds. The rate-distortion function that we study is equivalent to one defined the *information bottleneck* Tishby et al. [2000]. However, instead of using it as a basis for *optimization methods* as do Shwartz-Ziv and Tishby [2017], we develop tools to study *sample complexity* and arrive at concrete and novel results.

In this paper, we consider contexts in which an agent learns from an iid sequence of data pairs. We establish tight upper and lower bounds for the average regret and sample complexity that depend on the rate-distortion function. With these information-theoretic tools, we analyze two deep neural network data generating processes and quantify the number of samples required to arrive at a useful model. **We establish novel sample complexity bounds for ReLU neural network data generating processes that are roughly linear in the parameter count** (as opposed to linear in the product of parameter count and depth as in Harvey et al. [2017]). **For a multilayer process with arbitrary width but bounded sum of weights, we establish sample complexity bounds with only a quadratic depth dependence** as opposed to exponential or high-order polynomial dependence as in Bartlett et al. [2017] and Wei and Ma [2019] respectively. The **main contribution** of this paper is an **elegant and intuitive information-theoretic framework for analyzing machine learning problems**. We demonstrate its promise by deriving the two aforementioned theoretical results pertaining to deep neural networks which were not possible to show with existing analysis methods.

## 2   Prediction and Error

We begin by introducing the structure of data generating processes we consider in our framework and defining our notions of error and sample complexity.

## 2.1 Data Generating Process

We consider a stochastic process that generates a sequence $((X_t, Y_{t+1}) : t = 0, \ldots, T-1)$ of data pairs. We refer to each $X_t$ as an *input* and each $Y_{t+1}$ as an *output*. We define these and all other random variables we will consider with respect to a probability space $(\Omega, \mathbb{F}, \mathbb{P})$.

Elements of the sequence $(X_t : t = 0, \ldots, T-1)$ are independent and identically distributed. Denote the history of data generated through time $t$ by $H_t = (X_0, Y_1, \ldots, X_{t-1}, Y_t, X_t)$. We assume that each $(X_t, Y_{t+1})$ pair is exchangeable. As a result of de Finetti's theorem, there exists a latent variable $\mathcal{E}$ for which the pairs $(X_t, Y_{t+1})$ are iid conditioned on $\mathcal{E}$. Here, $\mathcal{E}$ (the environment) is a random function that specifies a conditional output distribution $\mathbb{P}(Y_{t+1} \in \cdot | \mathcal{E}, X_t = x) = \mathcal{E}(\cdot | x)$ for each input $x$. Initial uncertainty about $\mathcal{E}$ is expressed by the prior distribution $\mathbb{P}(\mathcal{E} \in \cdot)$.

## 2.2 Prediction

We consider an agent that predict the next response $Y_{t+1}$ given the history $H_t$. Rather than a point estimate, the agent provides as its prediction a probability distribution $P_t$ over possible responses. We characterize the agent in terms of a function $\pi$ for which $P_t = \pi(H_t)$.

We now introduce some notation for referring to particular predictions. We will generally use $P_t$ as a dummy variable – that is a generic prediction whose definition depends on context. We denote the prediction conditioned on the environment, which could only be produced by a prescient agent, by

$$P_t^* = \mathbb{P}(Y_{t+1} \in \cdot | \mathcal{E}, X_t) = \mathcal{E}(\cdot | X_t).$$

We will refer to this as the *target distribution* as it represents what an agent aims to learn. Finally, we use $\hat{P}_t$ to denote the posterior-predictive conditioned on $H_t$, which will turn out to be optimal for the objective we will define next.

$$\hat{P}_t = \mathbb{P}(Y_{t+1} \in \cdot | H_t).$$

## 2.3 Error

We assess the error of a prediction $P_t$ in terms of the KL-divergence relative to $P_t^*$:

$$\mathbf{d}_{\mathrm{KL}}(P_t^* \| P_t) = \int P_t^*(dy) \ln \frac{dP_t^*}{dP_t}(y).$$

This quantifies mismatch between the prediction $P_t$ and target distribution $P_t^*$. This notion of error is closely related to more common error notions from machine learning. When $Y$ takes values in a countable set, this notion of error is equivalent to expected cross-entropy loss. Meanwhile when $Y$ is continuous, this notion of error is upper and lower bounded by linear functions of mean-squared error under reasonable circumstances (refer to Appendix A).

# 3 Regret and Sample Complexity

We assess an agent's performance over duration $T$ in terms of the expected cumulative error

$$\mathcal{R}_\pi(T) = \mathbb{E}\left[\sum_{t=0}^{T-1} \mathbf{d}_{\mathrm{KL}}(P_t^* \| P_t)\right].$$

The focus of this paper is on understanding how well an *optimal* agent can perform, given particular data generating processes. We will use *regret* to refer to the optimal performance defined below.

**Definition 1. (optimal regret)** *For all $T \in \mathbb{Z}_+$, environments $\mathcal{E}$ and data generating processes $((X_0, Y_1), (X_1, Y_2), \ldots)$, the **optimal regret** is*

$$\mathcal{R}(T) := \inf_\pi \mathcal{R}_\pi(T).$$

We will also consider sample complexity, which we take to be the duration required to attain expected average error within some fraction $\epsilon \geq 0$ of an optimal uninformed prediction:

**Definition 2. (sample complexity)** *For all $\epsilon \geq 0$, environments $\mathcal{E}$ and data generating processes* $((X_0, Y_1), (X_1, Y_2), \ldots)$, *the **sample complexity** is*

$$T_\epsilon := \min\left\{T : \frac{\mathcal{R}(T)}{T} \leq \epsilon\right\}.$$

Note that this definition of sample complexity measures the number of samples necessary to achieve average *cumulative* error below $\epsilon$. Traditionally, generalization error quantifies the number of samples necessary to achieve out-of-sample error (such as $\mathbb{I}(Y_{T+1}; \mathcal{E}|H_T)$) below $\epsilon$. Due to a simple monotonicity result, our notion of sample complexity upper bounds the one for out-of-sample error i.e $\mathbb{I}(Y_{T+1}; \mathcal{E}|H_T) \leq \frac{\mathcal{R}(T)}{T}$.

### 3.1 Optimal Predictions

In this paper, we focus on how well an *optimal* agent performs, rather than on how to design practical agents that economize on memory and computation. Recall that an agent is characterized by a function $\pi$, which generates predictions $P_t = \pi(H_t, Z_t)$, where $Z_t$ represents algorithmic randomness. The following result establishes that the conditional distribution $\hat{P}_t = \mathbb{P}(Y_{t+1} \in \cdot | H_t)$ offers an optimal prediction.

**Theorem 3. (optimal prediction)** *For all $t \geq 0$,*

$$\mathbb{E}[\mathbf{d}_{\mathrm{KL}}(P_t^* \| \hat{P}_t) \mid H_t] = \inf_\pi \mathbb{E}[\mathbf{d}_{\mathrm{KL}}(P_t^* \| P_t) \mid H_t],$$

*where $P_t = \pi(H_t, Z_t)$.*

A proof may be found in Appendix B. A corollary that characterizes the performance shortfall of a *suboptimal* agent may also be found there. This corrolary opens future avenues regarding learning under misspecification or compute constraints. In the remainder of the paper we will study an agent that generates optimal predictions $P_t = \mathbb{P}(Y_{t+1} \in \cdot | H_t)$, as illustrated in Figure 1.

$$H_t = (X_0, Y_1, X_1, Y_2, \ldots, X_t) \longrightarrow \qquad \longrightarrow \hat{P}_t = \mathbb{P}(Y_{t+1} \in \cdot | H_t)$$

Figure 1: We consider an agent that, given a history $H_t$, generates an optimal prediction $\hat{P}_t$.

## 4 Information

We define concepts for quantifying uncertainty and the information gained from observations. The entropy $\mathbb{H}(\mathcal{E})$ of the environment quantifies the agent's initial degree of uncertainty in terms of the information required to identify $\mathcal{E}$. We will measure information in $nats$, each of which is equivalent to $1/\ln 2$ bits. For example, if $\mathcal{E}$ occupies a countable range $\Theta$ then $\mathbb{H}(\mathcal{E}) = -\sum_{\theta \in \Theta} \mathbb{P}(\mathcal{E} = \theta) \ln \mathbb{P}(\mathcal{E} = \theta)$. Uncertainty at time $t$ can be expressed in terms of the conditional entropy $\mathbb{H}(\mathcal{E}|H_t)$, which is the number of remaining nats after observing $H_t$. The mutual information $\mathbb{I}(\mathcal{E}; H_t) = \mathbb{H}(\mathcal{E}) - \mathbb{H}(\mathcal{E}|H_t)$ quantifies the information about $\mathcal{E}$ gained from $H_t$. Proofs for all lemmas in this section may be found in Appendix B

### 4.1 Learning from Errors

Each data pair $(X_t, Y_{t+1})$ provides $\mathbb{I}(\mathcal{E}; (X_t, Y_{t+1})|H_{t-1}, Y_t)$ nats of new information about the environment. By the chain rule of mutual information, this is the sum

$$\mathbb{I}(\mathcal{E}; (X_t, Y_{t+1})|H_{t-1}, Y_t) = \mathbb{I}(\mathcal{E}; X_t|H_{t-1}, Y_t) + \mathbb{I}(\mathcal{E}; Y_{t+1}|H_t)$$

of the information gained from $X_t$ and $Y_{t+1}$. The former term $\mathbb{I}(\mathcal{E}; X_t | H_{t-1}, Y_t)$ is equal to $0$ because $X_t$ is independent from both $\mathcal{E}$ and $(H_{t-1}, Y_t)$. The latter term $\mathbb{I}(\mathcal{E}; Y_{t+1} | H_t)$ can be thought of as the level of surprise experienced by the agent upon observing $Y_{t+1}$. Surprise is associated with prediction error, and the following result formalizes the equivalence between error and information.

**Lemma 4. (expected prediction error equals information gain)** *For all $t \in \mathbb{Z}_+$,*

$$\mathbb{E}[\mathbf{d}_{\mathrm{KL}}(P_t^* \| \hat{P}_t)] = \mathbb{I}(\mathcal{E}; Y_{t+1} | H_t),$$

*and $\mathcal{R}(t) = \mathbb{I}(\mathcal{E}; H_t)$.*

The agent's ability to predict tends to improve as it learns from experience. This is formalized by the following result, which establishes that expected prediction errors are monotonically nonincreasing.

**Lemma 5. (expected prediction error is monotonically nonincreasing)** *For all $t \in \mathbb{Z}_+$,*

$$\mathbb{E}[\mathbf{d}_{\mathrm{KL}}(P_t^* \| \hat{P}_t)] \geq \mathbb{E}[\mathbf{d}_{\mathrm{KL}}(P_{t+1}^* \| \hat{P}_{t+1})].$$

## 5  General Bounds via Rate-Distortion Theory

An *environment proxy* is a random variable $\tilde{\mathcal{E}}$ that provides information about the environment $\mathcal{E}$ but no additional information pertaining to inputs or outputs. In other words, $\tilde{\mathcal{E}} \perp (X, Y) | \mathcal{E}$. We will denote the set of environment proxies by $\tilde{\Theta}$. While an infinite amount of information must be acquired to identify the environment when $\mathbb{H}(\mathcal{E}) = \infty$, there can be a proxy $\tilde{\mathcal{E}}$ with $\mathbb{H}(\tilde{\mathcal{E}}) < \infty$ that enables accurate predictions. The minimal expected error attainable based on the proxy is achieved by a prediction $\tilde{P} = \mathbb{P}(Y \in \cdot | \tilde{\mathcal{E}}, X_t)$. This results in expected error $\mathbb{E}[\mathbf{d}_{\mathrm{KL}}(P^* \| \tilde{P})]$.

We will establish that the expected error $\mathbb{E}[\mathbf{d}_{\mathrm{KL}}(P^* \| \tilde{P})]$ equals the information gained, beyond that supplied by the proxy $\tilde{\mathcal{E}}$, about the environment $\mathcal{E}$ from observing $Y$. This is intuitive: more is learned from $Y$ if knowledge of $\mathcal{E}$ enables a better prediction of $Y$ than does $\tilde{\mathcal{E}}$. We quantify this information gain in terms of the difference $\mathbb{H}(Y | \tilde{\mathcal{E}}, X) - \mathbb{H}(Y | \mathcal{E}, X)$ between the uncertainty conditioned on $\tilde{\mathcal{E}}$ and that conditioned on $\mathcal{E}$. This is equal to the mutual information $\mathbb{I}(\mathcal{E}; Y | \tilde{\mathcal{E}}, X) = \mathbb{H}(Y | \tilde{\mathcal{E}}, X) - \mathbb{H}(Y | \mathcal{E}, X)$. The following result equates this with expected error.

**Lemma 6. (proxy error equals information gain)** *For all $\tilde{\mathcal{E}} \in \tilde{\Theta}$,*

$$\mathbb{E}[\mathbf{d}_{\mathrm{KL}}(P^* \| \tilde{P})] = \mathbb{I}(\mathcal{E}; Y | \tilde{\mathcal{E}}, X).$$

A proof may be found in Appendix C. $\mathbb{E}[\mathbf{d}_{\mathrm{KL}}(P^* \| \tilde{P})]$ is the *distortion* incurred by our estimate of $Y$ given only $\tilde{\mathcal{E}}$ as opposed to $\mathcal{E}$. For example, $\tilde{\mathcal{E}}$ may be a quantization or lossy compression of $\mathcal{E}$ an $\mathbb{E}[\mathbf{d}_{\mathrm{KL}}(P^* \| \tilde{P})]$ is measuring how inaccurate our prediction of $Y$ is under this compression $\tilde{\mathcal{E}}$.

Now we consider the following $\epsilon$-optimal set:

$$\tilde{\Theta}_\epsilon = \left\{ \tilde{\mathcal{E}} \in \tilde{\Theta} : \mathbb{E}[\mathbf{d}_{\mathrm{KL}}(P^* \| \tilde{P})] \leq \epsilon \right\},$$

$\tilde{\Theta}_\epsilon$ denotes the set of proxies that produce predictions that incur a *distortion* of no more than an $\epsilon$. One can imagine that for environments with simple input distributions, such as those constrained to a subspace or low-dimensional manifold, this set $\tilde{\Theta}_\epsilon$ will be much larger since the average error need only be below $\epsilon$ in a small subset of the domain.

With the *distortion* component of *rate-distortion* covered, the *rate* remains. The *rate* is $\mathbb{I}(\mathcal{E}; \tilde{\mathcal{E}})$, the amount of information about the environment conveyed by the proxy. For example, a finer quantization would result in a higher rate since $\tilde{\mathcal{E}}$ would capture $\mathcal{E}$ with more bits of precision. However, with this higher rate, the *distortion* incurred by $\tilde{\mathcal{E}}$ should in turn be *lower*. A higher fidelity compression should cost more bits but result in less distortion. The *rate-distortion function* formalizes this trade-off concretely:

**Definition 7.** *For all $\epsilon \geq 0$ and environments $\mathcal{E}$, The **rate-distortion function** w.r.t distortion function* $\mathbb{E}[\mathbf{d}_{\mathrm{KL}}(P^* \| \tilde{P})]$ *is*

$$\mathbb{H}_\epsilon(\mathcal{E}) := \inf_{\tilde{\mathcal{E}} \in \tilde{\Theta}_\epsilon} \mathbb{I}(\mathcal{E}; \tilde{\mathcal{E}}).$$

$\mathbb{H}_\epsilon$ characterizes the minimal amount of information that a proxy must convey about the environment in order to make $\epsilon$-accurate predictions. Even when $\mathbb{H}(\mathcal{E})$ is infinite and $\epsilon$ is small, $\mathbb{H}_\epsilon(\mathcal{E})$ can be manageable. Once again, if we consider an environment with a simple input distribution, the rate-distortion would be lower since we would be taking an infimum over a larger set $\tilde{\Theta}_\epsilon$. Depending on the structure of the input distribution and the environment, one could imagine that the reduction could be significant. As we will see in the following section, both the regret and sample complexity of learning scales with $\mathbb{H}_\epsilon(\mathcal{E})$.

## 5.1 Bounding Regret and Sample Complexity via Rate-Distortion

With rate-distortion in place, we produce tight bounds on regret and sample complexity. These bounds are very general, applying to *any* data generating process. The results upper and lower bound error and sample complexity in terms of rate-distortion. As such, for any particular data generating process, bounds can be produced by characterizing the associated rate-distortion function. Proofs for the following two theorems can be found in Appendix C.

The following result brackets the cumulative error of optimal predictions.

**Theorem 8. (rate-distortion regret bounds)** *For all $T$,*

$$\sup_{\epsilon \geq 0} \min\{\mathbb{H}_\epsilon(\mathcal{E}), \ \epsilon T\} \ \leq \ \mathcal{R}(T) \ \leq \ \inf_{\epsilon \geq 0}(\mathbb{H}_\epsilon(\mathcal{E}) + \epsilon T).$$

This upper bound is intuitive. Knowledge of a proxy $\tilde{\mathcal{E}} \in \tilde{\Theta}_\epsilon$ enables an agent to limit prediction error to $\epsilon$ per timestep. Getting to that level of prediction error requires $\mathbb{H}_\epsilon(\mathcal{E})$ nats, and therefore, that much cumulative error. Hence, $\mathcal{R}(T) \leq \mathbb{H}_\epsilon(\mathcal{E}) + \epsilon T$.

To motivate the lower bound, note that an agent requires $\mathbb{H}_\epsilon(\mathcal{E})$ nats to attain per timestep error within $\epsilon$. Obtaining those nats requires cumulative error at least $\mathbb{H}_\epsilon(\mathcal{E})$. So prior to obtaining that many nats, the agent *must* incur at least $\epsilon$ error per timestep, hence the $\epsilon T$ term in the minimum. Meanwhile, if at time $T$, the agent is able to produce predictions with error less than $\epsilon$ it means that it has already accumulated at least $\mathbb{H}_\epsilon(\mathcal{E})$ nats of information about $\mathcal{E}$ (error).

Sample complexity bounds follow almost immediately from Theorem 8.

**Theorem 9. (rate-distortion sample complexity bounds)** *For all $\epsilon \geq 0$,*

$$\frac{\mathbb{H}_\epsilon(\mathcal{E})}{\epsilon} \ \leq \ T_\epsilon \ \leq \ \inf_{\delta \in [0,\epsilon]} \left\lceil \frac{\mathbb{H}_{\epsilon-\delta}(\mathcal{E})}{\delta} \right\rceil \leq \left\lceil \frac{2\mathbb{H}_{\epsilon/2}(\mathcal{E})}{\epsilon} \right\rceil.$$

# 6 Deep Neural Network Environments

We now study the rate-distortion function of multilayer environments. As we established, sample complexity is governed by the rate-distortion function $\mathbb{H}_\epsilon(\mathcal{E})$. In this section, we will focus on characterizing this rate-distortion function, and hence the sample complexity, of two prototypical multilayer environments.

## 6.1 Prototypical Multilayer Environments

### 6.1.1 Prototypical Environment 1: (ReLU Neural Networks)

Prototypical Environment 1 mirrors the architecture of fully-connected feed-forward neural networks (refer to Figure 2 left). Let

$$U_1 = f_1(U_0) = \mathrm{ReLU}(A^{(1)}U_0 + b^{(1)}),$$

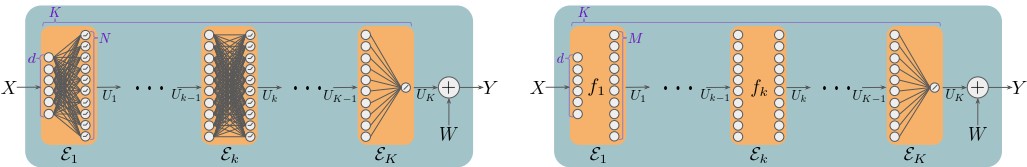

Figure 2: (Left) Prototypical environment 1, (Right) Prototypical environment 2.

where $A^{(1)} \in \Re^{N \times d}, b^{(1)} \in \Re^N$ both with independent elements each with variance $\frac{1}{d}$. For $k \in \{2, \ldots, K-1\}$, let

$$U_k = f_k(U_{k-1}) = \text{ReLU}(A^{(k)}U_{k-1} + b^{(k)}),$$

where $A^{(k)} \in \Re^{N \times N}, b^{(k)} \in \Re^N$, both with independent elements each with variance $\frac{1}{N}$. For the final layer, let

$$U_K = f_K(U_{K-1}) = A^{(K)\top}U_{K-1},$$

where $A^{(K)} \in \Re^N$ with independent elements, each with variance $\frac{1}{N}$. In this environment, $\mathcal{E}_k$ is identified by $(A^{(k)}, b^{(k)})$ for $k \in \{1, \ldots, K-1\}$ and $\mathcal{E}_K$ is identified by $A^{(K)}$. We will refer to this environment as the *deep ReLU network*.

### 6.1.2 Prototypical Environment 2: (Deep Nonparametric Networks)

Prototypical Environment 2 considers a deep nonparametric neural network (refer to Figure 2 right). For $k \in \{1, \ldots, K\}$, let

$$U_k = f_k(U_{k-1}) = \sqrt{c} \cdot \sum_{n=1}^{N} \delta_n^{(k)} \cdot \alpha_n^{(k)} \cdot g_n^{(k)}(U_{k-1}),$$

where $c \in \mathbb{Z}_{++}, \alpha^{(k)} = (\alpha_1^{(k)}, \ldots, \alpha_N^{(k)}) \sim \text{Dir}\left(N, \left[\frac{c}{N}, \ldots, \frac{c}{N}\right]\right)$, and $\delta_n^{(k)} \overset{iid}{\sim}$ Rademacher. For $k \in \{2, \ldots, K-1\}$, the deterministic basis functions $g_n^{(k)}$ are maps from $\Re^M \mapsto \Re^M$. $(g_n^{(1)} : n \in \{1, \ldots, N\})$ are maps from $\Re^d \mapsto \Re^M$ and $(g_n^{(K)} : n \in \{1, \ldots, N\})$ are maps from $\Re^M \mapsto \Re$. For regularity, we will assume that for all $n$ and $k$, the basis functions satisfy $\mathbb{E}[g_n^{(k)}(U_{k-1})^2] \leq 1$.

$\mathcal{E}_k$ is identified by $\delta^{(k)} \odot \alpha^{(k)}$. We will refer to this environment $(\mathcal{E}_{K:1})$ as the *deep nonparametric network*. For intractably large $N$, $f_k$ is effectively nonparametric. In this regime, parameter count becomes a vacuous description of the data generating process's complexity. However, the complexity is still controlled by the fact that $\sum_{n=1}^{N} \alpha_n = 1$ due to the Dirichlet prior. We will establish rate-distortion and sample complexity bounds that depend only logarithmically on $N$ and linearly in $c$.

A natural example of such an environment is one in which the basis functions are $g_n^{(k)}(X) = \text{ReLU}(\theta_{n,k}^\top X)$ where $\|\theta\|_2 = 1$ and $N$ may be prohibitively large i.e. exponential in the input dimension.

### 6.2 From Single to Multilayer Environments

We are now interested in analyzing the *depth* dependence of the rate-distortion and sample complexity of these environments. We will use $\mathcal{E}_{i:j}$ and $\tilde{\mathcal{E}}_{i:j}$ for $i \geq j$ to denote $(\mathcal{E}_i, \mathcal{E}_{i-1}, \ldots, \mathcal{E}_j)$ and $(\tilde{\mathcal{E}}_i, \tilde{\mathcal{E}}_{i-1}, \ldots, \tilde{\mathcal{E}}_j)$ respectively.

For *multilayer* environments, the error is cumbersome to reason about. Figure 3 (left) depicts the error incurred by using a multilayer proxy $\tilde{\mathcal{E}}_{K:1}$ to approximate multilayer environment $\mathcal{E}_{K:1}$. Evidently, it seems difficult to manage the error propagation through the layers of the environment. Many traditional lines of analysis struggle on this front and result in sample complexity bounds that are exponential in the depth Bartlett et al. [2017], Golowich et al. [2018]. These papers consider a worst-case reasoning under which an $\epsilon$ error between the first outputs $U_1$ and $\tilde{U}_1$ may blow up to a $\lambda^K \epsilon$ error when passed through remaining layers of the network (where $\lambda$ is a spectral radius).

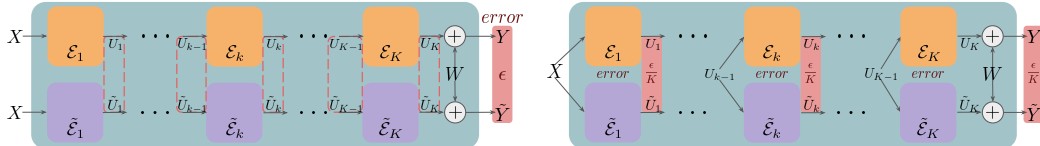

Figure 3: (Left) The error incurred by a multilayer proxy $\tilde{\mathcal{E}}_{K:1}$ measures the difference between true output $Y$ and the prediction $\tilde{Y}$ (shown in red box). This difference is the result of error that builds up through layers of the environment (denoted by red dotted outline). (Right) A much easier system to analyze is one in which we measure the incremental error at each stage of the multilayer environment. If each layer incurs an error of $\frac{\epsilon}{K}$, the total error incurred will be $\epsilon$.

It would be much simpler to instead analyze the incremental error incurred at each stage of the network. Figure 3 (right) depicts this. We consider that at each layer, we know the true input $U_{k-1}$ and simply measure the immediate error incurred at the output $U_k$ as opposed to the error incurred at the final output of the network $Y$.

Mathematically, the error incurred by the full system from using proxy $\tilde{\mathcal{E}}_{K:1}$ can be expressed as $\mathbb{I}(Y; \mathcal{E}_{K:1} | \tilde{\mathcal{E}}_{K:1}, X)$. By the chain rule, we have that

$$\mathbb{I}(Y; \mathcal{E}_{K:1} | \tilde{\mathcal{E}}_{K:1}, X) = \sum_{k=1}^{K} \mathbb{I}(Y; \mathcal{E}_k | \mathcal{E}_{K:k+1}, \tilde{\mathcal{E}}_{k:1}, X). \tag{1}$$

Therefore, the error incurred from layer $k$ can be expressed as $\mathbb{I}(Y; \mathcal{E}_k | \mathcal{E}_{K:k+1}, \tilde{\mathcal{E}}_{k:1}, X)$. This is cumbersome since we are not given the true input $U_{k-1}$ but rather an approximation from input $X$ and $\tilde{\mathcal{E}}_{k-1:1}$. Furthermore, we are measuring the error in the *final* output $Y$ as opposed to the *immediate* output $U_k$. It would be much more simple to analyze something like the following:

$$\sum_{k=1}^{K} \mathbb{I}(U_k + W; \mathcal{E}_k | \tilde{\mathcal{E}}_k, U_{k-1}), \tag{2}$$

where $W$ is independent 0-mean gaussian noise with variance $\sigma^2$ in each dimension. This sum is much easier to work with because the $k$th term only depends on $\mathcal{E}_k, \tilde{\mathcal{E}}_k, U_{k-1}$, and $U_k$. There is no inter-layer dependence.

The key insight is that in our two prototypical environments (and many others), something *akin* to the following will hold:

$$\mathbb{I}(Y; \mathcal{E}_k | \mathcal{E}_{K:k+1}, \tilde{\mathcal{E}}_{k:1}, X) \le \mathbb{I}(U_k + W; \mathcal{E}_k | \tilde{\mathcal{E}}_k, U_{k-1}). \tag{3}$$

As a result, the cumbersome sum 1 will be *upper bounded* by the nice sum 2.

The condition in inequality 3 involves two parts:

1. $\mathbb{I}(Y; \mathcal{E}_k | \mathcal{E}_{K:k+1}, \tilde{\mathcal{E}}_{k:1}, X) \le \mathbb{I}(Y; \mathcal{E}_k | \mathcal{E}_{K:k+1}, \tilde{\mathcal{E}}_k, U_{k-1})$
   - Conditioning on the true input $U_{k-1}$ provides more information about $\mathcal{E}_k$ than conditioning on an approximation $(\tilde{\mathcal{E}}_{k-1:1}, X)$.
2. $\mathbb{I}(Y; \mathcal{E}_k | \mathcal{E}_{K:k+1}, \tilde{\mathcal{E}}_k, U_{k-1}) \le \mathbb{I}(U_k + W; \mathcal{E}_k | \tilde{\mathcal{E}}_k, U_{k-1})$
   - The immediate output $U_k + W$ provides more information about $\mathcal{E}_k$ than the final output $Y$ does.

1) holds for all proxies of the form $\tilde{\mathcal{E}} = (\tilde{\mathcal{E}}_1, \ldots, \tilde{\mathcal{E}}_K)$ where $\tilde{\mathcal{E}}_i \perp \tilde{\mathcal{E}}_j$ for $i \ne j$. We prove this result explicitly in Lemma 19 in Appendix E. It is rather intuitive that the pristine data pair $(U_{k-1}, Y)$ would provide more information about $\mathcal{E}_k$ than $(U_0, \tilde{\mathcal{E}}_{k-1:1}, Y)$ would. We expect some information to be lost through the noisy reconstruction of $U_{k-1}$ from $U_0, \tilde{\mathcal{E}}_{k-1:1}$.

2) will not hold exactly for our two environments. $\mathbb{I}(Y; \mathcal{E}_k | \mathcal{E}_{K:k+1}, U_{k-1})$ will not always be $\le$ $\mathbb{I}(U_k + W, \mathcal{E}_k | \tilde{\mathcal{E}}_k, U_{k-1})$. However, $\mathbb{I}(Y; \mathcal{E}_k | \mathcal{E}_{K:k+1}, U_{k-1})$ will be $\le$ a very natural upper bound

of $\mathbb{I}(U_k + W, \mathcal{E}_k | \mathcal{E}_k, U_{k-1})$, which we detail in Lemma 20 of Appendix E. Intuitively $U_k + W$ will provide more information about $\mathcal{E}_k$ than $Y$ so long as in expectation, the layers $f_{k+1}, \ldots, f_K$ don't amplify the scale of the output. If they were to amplify the scale, then the signal to noise ratio of $Y$ would look more favorable than that of $U_k + W$, leading to potentially information.

Concretely, if

$$L = \sup_{x,y} \mathbb{E}\left[ \frac{\|f^{(k)}(x) - f^{(k)}(y)\|_2^2}{\|x - y\|_2^2} \bigg| x = x, y = y \right] \leq 1,$$

then we will have the desired result. The expectation here is taken over the randomness in the function $f^{(k)}$. For both of our prototypical data generating processes, this condition is met.

$L$ can be thought of as an expected squared Lipschitz constant of $f^{(k)}$. The set of random functions for which $L \leq 1$ is much broader than say 1-Lipschitz functions because of both the square and the expectation. For example, $f(x) = Ax$ is $\|A\|_2$-Lipschitz but if say cov $[A] = I_n$, then $L = 1$.

For multilayer processes for which we can establish a relationship akin to inequality 3, we have the following relationship between the rate-distortion function $\mathbb{H}_\epsilon(\mathcal{E}_{K:1})$ and the individual rate-distortion functions $\mathbb{H}_\epsilon(\mathcal{E}_k)$ (proof in Appendix E).

**Theorem 10. multilayer rate-distortion bound** *For all $K \in \mathbb{Z}_{++}$, $\sigma^2, \epsilon \geq 0$, if $\mathcal{E}_{K:1}$ is a multilayer environment such that for all $k \in \{1, \ldots, K\}$ and $\delta \geq 0$, there exist $\tilde{\mathcal{E}}_k$ s.t.*

$$\mathbb{I}(Y; \mathcal{E}_k | \mathcal{E}_{K:k+1}, \tilde{\mathcal{E}}_{k:1}, X) \leq \mathbb{I}(U_k + W; \mathcal{E}_k | \tilde{\mathcal{E}}_k, U_{k-1}) \leq \delta,$$

*where $W \sim \mathcal{N}(0, \sigma^2 I)$, then*

$$\mathbb{H}_\epsilon(\mathcal{E}_{K:1}) \leq \sum_{k=1}^{K} \mathbb{H}_{\frac{\epsilon}{K}}(\mathcal{E}_k).$$

### 6.3 Sample Complexity Bounds for Multilayer Environments

With Theorem 10 in place, we present the two main results (proofs can be found in Appendix F).

**Theorem 11. (relu neural network rate-distortion and sample complexity bound)** *For all $d, N, K \in \mathbb{Z}_{++}$ and $\sigma^2, \epsilon \geq 0$, if multilayer environment $\mathcal{E}_{K:1}$ is the deep ReLU network with input $X : \Omega \mapsto \Re^d$ s.t. $\mathbb{V}[X] = I_d$ and output $Y \sim \mathcal{N}(U_K, \sigma^2)$, then*

$$\mathbb{H}_\epsilon(\mathcal{E}_{K:1}) = \tilde{\mathcal{O}}\left( KN^2 + dN \right), \quad T_\epsilon = \tilde{\mathcal{O}}\left( \frac{KN^2 + dN}{\epsilon} \right).$$

While this is a parameter-count based bound in nature, it improves upon existing sample complexity results based on VC Dimension Bartlett et al. [1998],Harvey et al. [2017] which under our notation would be $\tilde{O}(K^2 N^2 + KdN)$. Note however that our result is average case while theirs holds with high probability. Regardless, the reduction in sample complexity by a factor of $K$ is notable.

For nonparametric deep networks, we have the following result.

**Theorem 12. (nonparametric network rate-distortion and sample complexity bound)** *For all $c, d, N, M, K \in \mathbb{Z}_{++}$, $\sigma^2, \epsilon \geq 0$, if multilayer environment $\mathcal{E}_{K:1}$ is the deep nonparametric network with input $X : \Omega \mapsto \Re^d$ and output $Y \sim \mathcal{N}(U_K, \sigma^2)$, then*

$$\mathbb{H}_\epsilon(\mathcal{E}_{K:1}) \leq \frac{cK^2 \ln(2N)}{2\sigma^2 \epsilon}, \quad T_\epsilon \leq \frac{2cK^2 \ln(2N)}{\sigma^2 \epsilon^2}.$$

The nonparametric network's sample complexity grows only quadratically in $K$ not exponentially or high-order polynomially as in Bartlett et al. [2017] and Wei and Ma [2019] respectively.

## 7 Closing Remarks

We have introduced a novel and elegant information-theoretic framework for analyzing the sample complexity of data generating processes. We demonstrate its usefulness by proving two new theoretical results that suggests that it is possible to learn efficiently from data generated by deep and nonparametric functions.

Beyond the scope of this paper, we believe that the flexibility and simplicity of our framework will allow for the analysis of machine learning systems such as semi-supervised learning, multitask learning, bandits and reinforcement learning. We also believe that many of the nuances of empirical deep learning such as batch-normalization, pooling, and structured input distributions can be analyzed through the average-case nature of information theory and powerful tools such as the data processing inequality.

Notably omitted from this paper is any analysis of practical algorithms for estimating the environment. We have assumed perfect Bayesian inference, while in practice, particular neural network architectures are used together with stochastic gradient descent. Whether practical algorithms of this sort can achieve sample complexity bounds similar to what we have established for our multi-layer environments remains an interesting subject for future research. Our discussion of suboptimal algorithms and misspecification in section 3.1 and Appendix B provide some starting points for such a pursuit.

## Acknowledgments and Disclosure of Funding

We thank Andrea Montari for suggesting that information-theoretic tools may be helpful to understanding the dependence of sample complexity on depth. Financial support from Army Research Office (ARO) grant W911NF2010055 and the National Science Foundation (NSF) Graduate Research Fellowships Program (GRFP) are gratefully acknowledged.

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
