# A  Relation to Other Notions of Error

## A.1  Connections to Cross-Entropy Loss

We establish that in the classification setting, our notion of error is equivalent to cross-entropy loss up to translations.

**Example 1. (cross-entropy loss)** *Suppose the set $\mathcal{Y}$ of responses is finite. Then,*

$$\mathbf{d}_{\mathrm{KL}}(P^*\|P) = \sum_{y\in\mathcal{Y}} P^*(y) \ln \frac{P^*(y)}{P(y)} = \sum_{y\in\mathcal{Y}} P^*(y) \ln P^*(y) - \sum_{y\in\mathcal{Y}} P^*(y) \ln P(y).$$

*The first term does not depend on $P$, so minimizing KL-divergence is equivalent to minimizing the final term,*

$$-\sum_{y\in\mathcal{Y}} P^*(y) \ln P(y) = -\mathbb{E}[\ln(P(Y_{t+1}))|\mathcal{E}, P, X],$$

*which is exactly the expected cross-entropy loss of $P$, as is commonly used to assess classifiers.*

## A.2  Connections to Mean-Squared Error

In the regression setting we first establish a direct link between KL-divergence and mean-squared error for the case in which $P_t^*$ and $P_t$ are Gaussian.

**Example 2. (Gaussian mean-squared error)** *Fix $\mu \in \Re$ and $\sigma^2 \in \Re_{++}$. Let $\mathbb{P}(Y_{t+1} \in \cdot | \mathcal{E}, X_t) \sim \mathcal{N}(\mu, \sigma^2)$. Consider a point prediction $\hat{\mu}_t$ that is determined by $H_t$ and a distributional prediction $P_t \sim \mathcal{N}(\hat{\mu}_t, \sigma^2)$. Then,*

$$\mathbf{d}_{\mathrm{KL}}(P_t^*\|P_t) = \frac{\mathbb{E}[(\mu - \hat{\mu}_t)^2|\mathcal{E}, H_t]}{2\sigma^2} = \frac{\mathbb{E}[(Y_{t+1} - \hat{\mu}_t)^2|\mathcal{E}, H_t] - \mathbb{E}[(Y_{t+1} - \mu_t)^2|\mathcal{E}, H_t]}{2\sigma^2}.$$

*Hence, KL-divergence grows monotonically the squared error $\mathbb{E}[(Y_{t+1} - \hat{\mu}_t)^2|\mathcal{E}, H_t]$. However, while the minimal squared error $\mathbb{E}[(Y_{t+1} - \mu)^2|\mathcal{E}, H_t] = \sigma^2$ that is attainable with full knowledge of the environment remains positive, the minimal KL-divergence, which is delivered by $P_t \sim \mathcal{N}(\mu, \sigma^2)$, is zero.*

*Now consider a distributional prediction $P_t \sim \mathcal{N}(\hat{\mu}_t, \hat{\sigma}_t^2)$, based on a variance estimate $\hat{\sigma}_t^2 \neq \sigma^2$. Then,*

$$\mathbf{d}_{\mathrm{KL}}(P_t^*\|P_t) = \frac{\mathbb{E}[(\mu - \hat{\mu}_t)^2|\mathcal{E}, H_t]}{2\hat{\sigma}_t^2} + \frac{1}{2}\left(\frac{\sigma^2}{\hat{\sigma}_t^2} - 1 - \ln\frac{\sigma^2}{\hat{\sigma}_t^2}\right).$$

*Consider optimizing the choice of $\hat{\sigma}_t^2$ given $H_t$:*

$$\min_{\hat{\sigma}_t^2} \mathbb{E}[\mathbf{d}_{\mathrm{KL}}(P_t^*\|P_t)|H_t].$$

*The minimum is attained by*

$$\hat{\sigma}_t^2 = \underbrace{\sigma^2}_{\text{aleatoric}} + \underbrace{\mathbb{E}[(\mu - \mathbb{E}[\mu|H_t])^2|H_t]}_{\text{epistemic}} + \underbrace{\mathbb{E}[(\mathbb{E}[\mu|H_t] - \hat{\mu}_t)^2|H_t]}_{bias},$$

*which differs from $\sigma^2$. While $\sigma^2$ characterizes aleatoric uncertainty, the incremental variance $\hat{\sigma}_t^2 - \sigma^2$ accounts for epistemic uncertainty and bias.*

Now, for $P_t^*$ and $P_t$ that are not Gaussian, we have the following upper bound:

**Lemma 13.** *For all $t \in \mathbb{Z}_+$, if $\hat{\mu}_t = \int_{y\in\mathcal{Y}} y \, dP_t(y)$, then*

$$\mathbb{E}[\mathbf{d}_{\mathrm{KL}}(P_t^*\|P_t)] \leq \frac{1}{2}\ln\left(1 + \frac{\mathbb{E}\left[(Y_{t+1} - \hat{\mu}_t)^2\right]}{\sigma^2}\right).$$

Therefore, decreasing the mean squared error will always decrease the expected KL-divergence. A corresponding lower bound holds for data generating processes for which $Y_{t+1}$ satisfies a certain subgaussian condition:

**Lemma 14.** *For all $t \in \mathbb{Z}_+$, let $\hat{\mu}_t = \int_{y \in \mathcal{Y}} y \, dP_t(y)$. If $P_t(Y_{t+1} \in \cdot)$ is $\delta_t^2$ -subgaussian conditioned on $H_t$ w.p 1, then*

$$\mathbb{E}\left[\mathbf{d}_{\mathrm{KL}}(P_t^* \| P_t)\right] \geq \frac{\mathbb{E}\left[(Y_{t+1} - \hat{\mu}_t)^2\right]}{\delta_t^2}$$

Therefore, for data generating processes that obey this subgaussian condition, we have both upper and lower bounds for expected KL divergence in terms of mean-squared error.

## B Proof of Information and Error Results

Here, we provide a proof of Theorem 3.

**Theorem 3. (optimal prediction)** *For all $t \geq 0$,*

$$\mathbb{E}[\mathbf{d}_{\mathrm{KL}}(P_t^* \| \hat{P}_t) \mid H_t] = \inf_\pi \mathbb{E}[\mathbf{d}_{\mathrm{KL}}(P_t^* \| P_t) \mid H_t],$$

*where $P_t = \pi(H_t, Z_t)$.*

*Proof.* Let $\hat{P}_t = \mathbb{P}(Y_{t+1} \in \cdot | H_t)$. By Gibbs' inequality,

$$\inf_{P_t} \mathbf{d}_{\mathrm{KL}}(\hat{P}_t \| P_t) = \mathbf{d}_{\mathrm{KL}}(\hat{P}_t \| \hat{P}_t) = 0.$$

Let $P_t^* = \mathbb{P}(Y_{t+1} \in \cdot | \mathcal{E}, X_t)$. Then, for all $P_t$,

$$
\begin{aligned}
\mathbf{d}_{\mathrm{KL}}(P_t^* \| P_t) =& \mathbb{E}\left[\ln \frac{dP_t^*}{dP_t}(Y_{t+1}) \Big| \mathcal{E}, H_t\right] \\
=& \mathbb{E}\left[\ln dP_t^*(Y_{t+1}) | \mathcal{E}, H_t\right] - \mathbb{E}\left[\ln dP_t(Y_{t+1}) | \mathcal{E}, H_t\right] \\
=& \mathbb{E}\left[\ln dP_t^*(Y_{t+1}) | \mathcal{E}, H_t\right] - \mathbb{E}\left[\ln d\hat{P}_t(Y_{t+1}) | \mathcal{E}, H_t\right] \\
& + \mathbb{E}\left[\ln d\hat{P}_t(Y_{t+1}) | \mathcal{E}, H_t\right] - \mathbb{E}\left[\ln dP_t(Y_{t+1}) | \mathcal{E}, H_t\right] \\
=& \mathbb{E}\left[\ln \frac{dP_t^*}{d\hat{P}_t}(Y_{t+1}) \Big| \mathcal{E}, H_t\right] + \mathbb{E}\left[\ln \frac{d\hat{P}_t}{dP_t}(Y_{t+1}) \Big| \mathcal{E}, H_t\right] \\
=& \mathbf{d}_{\mathrm{KL}}(P_t^* \| \hat{P}_t) + \mathbb{E}\left[\ln \frac{d\hat{P}_t}{dP_t}(Y_{t+1}) \Big| \mathcal{E}, H_t\right].
\end{aligned}
$$

It follows that

$$
\begin{aligned}
\inf_\pi \mathbb{E}[\mathbf{d}_{\mathrm{KL}}(P_t^* \| P_t) | H_t] =& \inf_\pi \mathbb{E}\left[\mathbf{d}_{\mathrm{KL}}(P_t^* \| \hat{P}_t) + \mathbb{E}\left[\ln \frac{d\hat{P}_t}{dP_t}(Y_{t+1}) \Big| \mathcal{E}, H_t\right] \Big| H_t\right] \\
=& \mathbb{E}[\mathbf{d}_{\mathrm{KL}}(P_t^* \| \hat{P}_t) | H_t] + \inf_\pi \mathbb{E}[\mathbf{d}_{\mathrm{KL}}(\hat{P}_t \| P_t) | H_t] \\
=& \mathbb{E}[\mathbf{d}_{\mathrm{KL}}(P_t^* \| \hat{P}_t) | H_t] + \mathbb{E}[\mathbf{d}_{\mathrm{KL}}(\hat{P}_t \| \hat{P}_t) | H_t] \\
=& \mathbb{E}[\mathbf{d}_{\mathrm{KL}}(P_t^* \| \hat{P}_t) | H_t].
\end{aligned}
$$

$\square$

A direct corollary of this result is the following shortfall quantification for a misspecified/suboptimal prediction.

**Corollary 15. (misspecified/suboptimal prediction)** *For all $t \geq 0$ and $P_t = \pi(H_t, Z_t)$,*

$$\mathbb{E}[\mathbf{d}_{\mathrm{KL}}(P_t^* \| P_t) \mid H_t] = \mathbb{E}[\mathbf{d}_{\mathrm{KL}}(P_t^* \| \hat{P}_t) \mid H_t] + \mathbb{E}[\mathbf{d}_{\mathrm{KL}}(\hat{P}_t \| P_t) \mid H_t].$$

An interpretation of the above result is that the error of a suboptimal agent with prediction $P_t$ is the sum of the error of an *optimal* agent $\hat{P}_t$ and the expected KL divergence between $\hat{P}_t$ and $P_t$. Since all the results in this paper bound the first term, in order to understand the behavior of a suboptimal agent, techniques to bound the second term would be required.

Next, we provide a proof for Lemma 4.

**Lemma 4. (expected prediction error equals information gain)** *For all $t \in \mathbb{Z}_+$,*

$$\mathbb{E}[\mathbf{d}_{\mathrm{KL}}(P_t^*\|\hat{P}_t)] = \mathbb{I}(\mathcal{E}; Y_{t+1}|H_t),$$

*and $\mathcal{R}(t) = \mathbb{I}(\mathcal{E}; H_t)$.*

*Proof.* It is well known that the mutual information $\mathbb{I}(A; B)$ between random variables $A$ and $B$ can be expressed in terms of the expected KL-divergence $\mathbb{I}(A; B) = \mathbb{E}[\mathbf{d}_{\mathrm{KL}}(\mathbb{P}(A \in \cdot|B)\|\mathbb{P}(A \in \cdot))]$. It follows that

$$\begin{aligned}
\mathbb{I}(Y_{t+1}; \mathcal{E}|H_t) &= \mathbb{E}[\mathbf{d}_{\mathrm{KL}}(\mathbb{P}(Y_{t+1} \in \cdot|\mathcal{E}, H_t) \| \mathbb{P}(Y_{t+1} \in \cdot|H_t))] \\
&\overset{(a)}{=} \mathbb{E}[\mathbf{d}_{\mathrm{KL}}(\mathbb{P}(Y_{t+1} \in \cdot|\mathcal{E}, X_t) \| \mathbb{P}(Y_{t+1} \in \cdot|H_t))] \\
&= \mathbb{E}[\mathbf{d}_{\mathrm{KL}}(P_t^*\|\hat{P}_t)],
\end{aligned}$$

where $(a)$ follows from the fact that $Y_{t+1} \perp H_t|(\mathcal{E}, X_t)$. We then have

$$\mathcal{R}(T) = \mathbb{E}\left[\sum_{t=0}^{T-1} \mathbf{d}_{\mathrm{KL}}(P_t^*\|\hat{P}_t)\right] = \sum_{t=0}^{T-1} \mathbb{I}(Y_{t+1}; \mathcal{E}|H_t) \overset{a}{=} \mathbb{I}(\mathcal{E}; H_T),$$

where $(a)$ follows from the chain rule of mutual information. $\qquad\square$

Lastly, we provide a proof for Lemma 5.

**Lemma 5. (expected prediction error is monotonically nonincreasing)** *For all $t \in \mathbb{Z}_+$,*

$$\mathbb{E}[\mathbf{d}_{\mathrm{KL}}(P_t^*\|\hat{P}_t)] \geq \mathbb{E}[\mathbf{d}_{\mathrm{KL}}(P_{t+1}^*\|\hat{P}_{t+1})].$$

*Proof.* We have

$$\begin{aligned}
\mathbb{E}[\mathbf{d}_{\mathrm{KL}}(P_{t+1}^*\|\hat{P}_{t+1})] &\overset{(a)}{=} \mathbb{I}(\mathcal{E}; Y_{t+2}|H_{t+1}) \\
&= \mathbf{h}(Y_{t+2}|H_{t+1}) - \mathbf{h}(Y_{t+2}|\mathcal{E}, H_{t+1}) \\
&\overset{(b)}{=} \mathbf{h}(Y_{t+2}|H_{t+1}) - \mathbf{h}(Y_{t+2}|\mathcal{E}, H_{t-1}, Y_t, X_{t+1}) \\
&\overset{(c)}{\leq} \mathbf{h}(Y_{t+2}|H_{t-1}, Y_t, X_{t+1}) - \mathbf{h}(Y_{t+2}|\mathcal{E}, H_{t-1}, Y_t, X_{t+1}) \\
&= \mathbb{I}(\mathcal{E}; Y_{t+2}|H_{t-1}, Y_t, X_{t+1}) \\
&\overset{(d)}{=} \mathbb{I}(\mathcal{E}; Y_{t+1}|H_{t-1}, Y_t, X_t) \\
&= \mathbb{I}(\mathcal{E}; Y_{t+1}|H_t) \\
&\overset{(e)}{=} \mathbb{E}[\mathbf{d}_{\mathrm{KL}}(P_t^*\|\hat{P}_t)],
\end{aligned}$$

where $(a)$ follows from Lemma 4, $(b)$ follows from the the fact that $Y_{t+2} \perp (X_t, Y_{t+1})|(\mathcal{E}, X_{t+1})$, $(c)$ follows from the fact that conditioning reduces differential entropy, $(d)$ follows from the fact that $(X_t, Y_{t+1})$ and $(X_{t+1}, Y_{t+2})$ are independent and identically distributed conditioned on $(H_{t-1}, Y_t)$, and $(e)$ follows from the equivalence between mutual information and expected KL-divergence. $\quad\square$

## C  Proofs of Regret and Sample Complexity Results

In this section, we prove results pertaining to bounding optimal regret and sample complexity in terms of rate-distortion.

**Lemma 6. (proxy error equals information gain)** *For all $\tilde{\mathcal{E}} \in \tilde{\Theta}$,*

$$\mathbb{E}[\mathbf{d}_{\mathrm{KL}}(P^*\|\tilde{P})] = \mathbb{I}(\mathcal{E};Y|\tilde{\mathcal{E}},X).$$

*Proof.* It is well known that the mutual information $\mathbb{I}(A;B)$ between random variables $A$ and $B$ can be expressed in terms of the expected KL-divergence $\mathbb{I}(A;B) = \mathbb{E}[\mathbf{d}_{\mathrm{KL}}(\mathbb{P}(A \in \cdot|B)\|\mathbb{P}(A \in \cdot))]$. We therefore have

$$\begin{aligned}
\mathbb{I}(\mathcal{E};Y|\tilde{\mathcal{E}},X) &= \mathbb{E}[\mathbf{d}_{\mathrm{KL}}(\mathbb{P}(Y \in \cdot|\mathcal{E},\tilde{\mathcal{E}},X) \,\|\, \mathbb{P}(Y \in \cdot|\tilde{\mathcal{E}},X))] \\
&= \mathbb{E}[\mathbf{d}_{\mathrm{KL}}(\mathbb{P}(Y \in \cdot|\mathcal{E},X) \,\|\, \mathbb{P}(Y \in \cdot|\tilde{\mathcal{E}},X))] \\
&= \mathbb{E}[\mathbf{d}_{\mathrm{KL}}(P^*\|\tilde{P})],
\end{aligned}$$

where the second equation follows from the fact that $(X,Y) \perp \tilde{\mathcal{E}}|\mathcal{E}$. $\qquad\square$

**Theorem 8. (rate-distortion regret bounds)** *For all $T$,*

$$\sup_{\epsilon \geq 0} \min\{\mathbb{H}_\epsilon(\mathcal{E}),\ \epsilon T\} \ \leq \ \mathcal{R}(T) \ \leq \ \inf_{\epsilon \geq 0}(\mathbb{H}_\epsilon(\mathcal{E}) + \epsilon T).$$

*Proof.* We begin by establishing the upper bound.

$$\begin{aligned}
\mathcal{R}(T) &= \sum_{t=0}^{T-1} \mathbb{E}\left[\mathbf{d}_{\mathrm{KL}}(P_T^*\|\hat{P}_t)\right] \\
&\overset{(a)}{=} \sum_{t=0}^{T-1} \mathbb{I}(Y_{t+1};\mathcal{E}|H_t) \\
&= \sum_{t=0}^{T-1} \mathbb{I}(Y_{t+1};\mathcal{E},\tilde{\mathcal{E}}|H_t) \\
&\overset{(b)}{=} \sum_{t=0}^{T-1} \mathbb{I}(Y_{t+1};\tilde{\mathcal{E}}|H_t) + \mathbb{I}(Y_{t+1};\mathcal{E}|\tilde{\mathcal{E}},H_t) \\
&\overset{(c)}{\leq} \sum_{t=0}^{T-1} \mathbb{I}(Y_{t+1};\tilde{\mathcal{E}}|H_t) + \mathbb{I}(Y_{t+1};\mathcal{E}|\tilde{\mathcal{E}},H_t) \\
&\overset{(d)}{=} \mathbb{I}(H_T;\tilde{\mathcal{E}}) + \sum_{t=0}^{T-1} \mathbb{I}(Y_{t+1};\mathcal{E}|\tilde{\mathcal{E}},H_t) \\
&\overset{(e)}{\leq} \mathbb{I}(H_T;\tilde{\mathcal{E}}) + \sum_{t=0}^{T-1} \mathbb{I}(Y_{t+1};\mathcal{E}|\tilde{\mathcal{E}},X_t) \\
&\overset{(f)}{\leq} \mathbb{I}(H_T;\tilde{\mathcal{E}}) + \epsilon T \\
&\overset{(g)}{\leq} \mathbb{I}(\mathcal{E};\tilde{\mathcal{E}}) + \epsilon T
\end{aligned}$$

where (a) follows from Lemma 4, (b) follows from the chain rule of mutual information, (c) follows from the chain rule of mutual information, (d) follows from the facts that $\mathbf{h}(Y_{t+1}|\tilde{\mathcal{E}},H_t) \leq \mathbf{h}(Y_{t+1}|\tilde{\mathcal{E}},X_t)$ and $\mathbf{h}(Y_{t+1}|\mathcal{E},H_t) = \mathbf{h}(Y_{t+1}|\mathcal{E},X_t)$, (e) holds for any $\tilde{\mathcal{E}} \in \tilde{\Theta}_\epsilon$, and (f) follows from the data processing inequality. Since the above inequality holds for all $\epsilon \geq 0$ and $\tilde{\mathcal{E}} \in \tilde{\Theta}_\epsilon$, the result follows.

Next, we establish the lower bound. Fix $T \in \mathbb{Z}_+$. Let $\tilde{\mathcal{E}} = (\tilde{H}_{T-2},\tilde{Y}_{T-1})$ be independent from but distributed identically with $(H_{T-2},Y_{T-1})$, conditioned on $\mathcal{E}$. In other words, $\tilde{\mathcal{E}} \perp (H_{T-2},Y_{T-1})|\mathcal{E}$ and $\mathbb{P}(\tilde{\mathcal{E}} \in \cdot|\mathcal{E}) = \mathbb{P}((H_{T-2},Y_{T-1}) \in \cdot|\mathcal{E})$. This implies that $\mathbb{P}((\mathcal{E},\tilde{H}_{T-2},\tilde{Y}_{T-1},X_{T-1},Y_T) \in \cdot) = \mathbb{P}((\mathcal{E},H_{T-2},Y_{T-1},X_{T-1},Y_T) \in \cdot)$, and therefore, $\mathbb{I}(\mathcal{E};Y_T|H_{T-1}) = \mathbb{I}(\mathcal{E};Y_T|\tilde{\mathcal{E}},X_{T-1})$.

Fix $\epsilon \geq 0$. If $\mathcal{R}(T) < \mathbb{H}_\epsilon(\mathcal{E})$ then,

$$
\begin{aligned}
\mathcal{R}(T) &\overset{(a)}{=} \mathbb{I}(\mathcal{E}; H_T) \\
&\overset{(b)}{=} \sum_{t=0}^{T-1} \mathbb{I}(\mathcal{E}; Y_{t+1}|H_t) \\
&\overset{(c)}{\geq} \mathbb{I}(\mathcal{E}; Y_T|H_{T-1})T \\
&= \mathbb{I}(\mathcal{E}; Y_T|\tilde{\mathcal{E}}, X_{T-1})T \\
&\overset{(d)}{=} \mathbb{E}[\mathbf{d}_{\mathrm{KL}}(P_{T-1}^* \| \tilde{P}_{T-1})]T \\
&\overset{(e)}{>} \epsilon T,
\end{aligned}
$$

where (a) follows from Lemma 4, (b) follows from the chain rule of mutual information, (c) follows from Lemma 5, (d) follows from Lemma 6, and (e) follows from the fact that $\tilde{\mathcal{E}} \notin \tilde{\Theta}_\epsilon$ because $\mathcal{R}(T) = \mathbb{I}(\mathcal{E}; H_T) = \mathbb{I}(\mathcal{E}; \tilde{\mathcal{E}}) < H_\epsilon(\mathcal{E})$. Therefore,

$$
\mathcal{R}(T) \geq \min\{\mathbb{H}_\epsilon(\mathcal{E}), \ \epsilon T\}.
$$

Since this holds for any $\epsilon \geq 0$, the result follows. □

**Theorem 9. (rate-distortion sample complexity bounds)** *For all $\epsilon \geq 0$,*

$$
\frac{\mathbb{H}_\epsilon(\mathcal{E})}{\epsilon} \ \leq \ T_\epsilon \ \leq \ \inf_{\delta \in [0,\epsilon]} \left\lceil \frac{\mathbb{H}_{\epsilon-\delta}(\mathcal{E})}{\delta} \right\rceil \leq \left\lceil \frac{2\mathbb{H}_{\epsilon/2}(\mathcal{E})}{\epsilon} \right\rceil.
$$

*Proof.* We begin by showing the upper bound. Fix $\epsilon \geq 0$ and $\delta \in [0, \epsilon]$. Let

$$
T = \left\lceil \frac{\mathbb{H}_{\epsilon-\delta}(\mathcal{E})}{\delta} \right\rceil,
$$

so that $\mathbb{H}_{\epsilon-\delta}(\mathcal{E}) \leq \delta T$. We have that:

$$
\begin{aligned}
\mathcal{R}(T) &\overset{(a)}{\leq} \mathbb{H}_{\epsilon-\delta}(\mathcal{E}) + (\epsilon - \delta)T \\
&\overset{(b)}{\leq} \delta T + (\epsilon - \delta)T \\
&= \epsilon T,
\end{aligned}
$$

where $(a)$ follows from the upper bound of Theorem 8 and $(b)$ follows from our choice of $T$. Since $T_\epsilon = \min\{T : \mathcal{R}(T) \leq \epsilon \mathcal{R}(1)T\}$, it follows that $T \geq T_\epsilon$. Since the above holds for arbitrary $\delta \in [0, \epsilon]$, the result follows.

We now show the lower bound. Fix $\epsilon \geq 0$. By the definition of $T_\epsilon$, we have

$$
\mathcal{R}(T_\epsilon) \leq \epsilon T_\epsilon.
$$

In the proof of the lower bound in Theorem 8, we show that for all $\epsilon \geq 0$, $\mathcal{R}(T) < \mathbb{H}_\epsilon(\mathcal{E}) \implies \mathcal{R}(T) > \epsilon T$. Therefore, using the contrapositive and the above definition of $T_\epsilon$, we have that $\mathbb{H}_\epsilon(\mathcal{E}) \leq \mathcal{R}(T_\epsilon)$ and therefore

$$
\mathbb{H}_\epsilon(\mathcal{E}) \leq \mathcal{R}(T_\epsilon) \leq \epsilon T_\epsilon.
$$

The result follows. □

## D  Proofs of Single-Layer Rate-Distortion Bounds

We have established in Theorem 10 that the rate-distortion function for a multilayer environment is a sum of the rate-distortion functions for each single-layer environment under tolerance $\frac{\epsilon}{K}$. In this section, we show the following rate-distortion bounds for a single layer of the deep ReLU network and the deep nonparametric network.

We begin by stating a well known result from information theory:

**Lemma 16. (maximum differential entropy)** *For all random vectors $X : \Omega \mapsto \Re^d$ with covariance $K$,*

$$\mathbf{h}(X) \le \frac{1}{2} \ln \left( (2\pi e)^d |K| \right),$$

*with equality iff $Pr(X \in \cdot) \sim \mathcal{N}(\mu, K)$ for some $\mu \in \Re^d$.*

*Proof.* Follows from Theorems 8.6.3 and 8.6.5 of Cover and Thomas [2006]. $\square$

**Theorem 17. (single-layer relu neural network rate-distortion bound)** *For all $N, k \in \mathbb{Z}_{++}$ and $\sigma^2, \epsilon \ge 0$, if $\mathcal{E}_k$ is identified by $(A, b)$ where $A : \Omega \mapsto \Re^{N \times N}, b : \Omega \mapsto \Re^N$ both consist of independent elements each with variance $\frac{1}{N}$, $X : \Omega \mapsto \Re^N$ is a random vector with covariance $I_N$, and $Y \sim \mathcal{N}(\text{ReLU}(AX + b), \sigma^2 I_N)$, then*

$$\mathbb{H}_\epsilon(\mathcal{E}_k) = \mathcal{O}\left( N^2 \ln\left( \frac{N}{2\sigma^2 \epsilon} \right) \right).$$

*Proof.* We use $A_i, b_i$ to denote the $i$th rows of $A, b$ respectively. Let $\tilde{\mathcal{E}}_k = (\tilde{A}, \tilde{b})$ where $\tilde{A} = A + V$ where $V \perp A$ and $V \sim \mathcal{N}(0, \delta^2 I_d)$ and likewise $\tilde{b} = b + Z$ where $Z \perp b$ and $Z \sim \mathcal{N}(0, \delta^2)$. $\delta^2 = \frac{\sigma^2 \left( e^{\frac{2\epsilon}{N}} - 1 \right)}{N+1}$. We have that

$$
\begin{aligned}
\mathbb{I}(\mathcal{E}_k; \tilde{\mathcal{E}}_k) &= \mathbb{I}(\mathcal{E}_k; \tilde{\mathcal{E}}_k) \\
&= \mathbb{I}(A; \tilde{A}) + \mathbb{I}(b; \tilde{b}) \\
&\le \mathbf{h}(\tilde{A}) - \mathbf{h}(\tilde{A}|A) + \mathbb{I}(b; \tilde{b}) \\
&= \frac{N^2}{2} \ln\left( 2\pi e \left( \delta^2 + \frac{1}{N} \right) \right) - \mathbf{h}(V) + \mathbb{I}(b; \tilde{b}) \\
&= \frac{N^2}{2} \ln\left( 1 + \frac{1}{N\delta^2} \right) + \frac{N}{2} \ln\left( 1 + \frac{1}{N\delta^2} \right) \\
&\le \frac{N(N+1)}{2} \ln\left( 1 + \frac{1}{N\delta^2} \right) \\
&= \mathcal{O}\left( N^2 \ln\left( 1 + \frac{1}{N\delta^2} \right) \right) \\
&= \mathcal{O}\left( N^2 \ln\left( 1 + \frac{1}{\sigma^2 \left( e^{\frac{2\epsilon}{N}} - 1 \right)} \right) \right) \\
&= \mathcal{O}\left( N^2 \ln\left( \frac{N}{2\sigma^2 \epsilon} \right) \right).
\end{aligned}
$$

We now verify that our choice of $\tilde{\mathcal{E}}_k$ satisfies the distortion constraint:

$$
\begin{aligned}
\mathbb{I}(Y; \mathcal{E}_k|\tilde{\mathcal{E}}_k, X) &= \mathbb{I}(Y; A, b|\tilde{A}, \tilde{b}, S, X) \\
&= \mathbf{h}(Y|\tilde{A}, \tilde{b}, S, X) - \mathbf{h}(Y|A, b, S, X) \\
&= N\left(\mathbf{h}(Y_i|\tilde{A}, \tilde{b}, S, X) - \mathbf{h}(Y_i|A, b, S, X)\right) \\
&\overset{(a)}{=} N\left(\mathbf{h}(Y_i|\tilde{A}_i, \tilde{b}_i, S_i, X) - \mathbf{h}(W_i)\right) \\
&= N\left(\mathbf{h}(Y_i - \mathrm{ReLU}(\tilde{A}_i^\top X + \tilde{b}_i)|\tilde{\theta}_i, \tilde{b}_i, S_i, X) - \mathbf{h}(W_i)\right) \\
&\overset{(b)}{\leq} N\left(\mathbf{h}(Y_i - \mathrm{ReLU}(\tilde{A}_i^\top X + \tilde{b}_i)|X) - \mathbf{h}(W_i)\right) \\
&\overset{(c)}{\leq} \mathbb{E}\left[N\left(\frac{1}{2}\ln\left(2\pi e\left(\sigma^2 + \mathbb{V}[\mathrm{ReLU}(A_i^\top X + b_i) - \mathrm{ReLU}(\tilde{A}_i^\top X + \tilde{b}_i)|X]\right)\right) - \mathbf{h}(W_i)\right)\right] \\
&\overset{(d)}{\leq} \frac{N}{2}\ln\left(1 + \frac{\mathbb{V}[\mathrm{ReLU}(A_i^\top X + b_i) - \mathrm{ReLU}(\tilde{A}_i^\top X + \tilde{b}_i)]}{\sigma^2}\right) \\
&\overset{(e)}{\leq} \frac{N}{2}\ln\left(1 + \frac{\mathbb{V}[A_i^\top X + b_i - \tilde{A}_i^\top X - \tilde{b}_i]}{\sigma^2}\right) \\
&= \frac{N}{2}\ln\left(1 + \frac{(N+1)\delta^2}{\sigma^2}\right) \\
&= \epsilon
\end{aligned}
$$

where in $(a)$, $W_i \sim \mathcal{N}(0, \sigma^2)$, $(b)$ follows from the fact that conditioning reduces entropy, $(c)$ follows from Lemma 16, $(d)$ follows from Jensen's Inequality, and $(e)$ follows from the fact that for all $x, y \in \Re$, $(\mathrm{ReLU}(x) - \mathrm{ReLU}(y))^2 \leq (x - y)^2$. $\qquad\square$

**Theorem 18. (single-layer nonparametric rate-distortion bound)** *For all $N, M, c \in \mathbb{Z}_{++}$ and $\sigma^2, \epsilon \geq 0$, if $\mathcal{E}$ is identified by $\alpha$ where $\alpha \sim Dir\left(N, \left[\frac{c}{N}, \ldots, \frac{c}{N}\right]\right)$ and*

$$
f(X) = \sqrt{c} \cdot \sum_{n=1}^{N} \alpha_n g_n(X)
$$

*for deterministic basis functions $(g_1, \ldots, g_N)$ mapping $\Re^M \mapsto \Re^M$ that satisfy $\mathbb{E}[g_n(X)^\top g_n(X)] \leq 1$ for random vector $X : \Omega \mapsto \Re^N$, and $Y \sim \mathcal{N}(f(X), \sigma^2 I_M)$, then*

$$
\mathbb{H}_\epsilon(\mathcal{E}) \leq \frac{c\ln(2N)}{2\sigma^2\epsilon}.
$$

*Proof.* Let $\tilde{\mathcal{E}} = \frac{\sqrt{c}}{r}\sum_{i=1}^{r} h_i$ for $r = \frac{c}{2\sigma^2\epsilon}$ and where for all $i \in \{1, \ldots, r\}$,

$$
h_i \overset{iid}{\sim} \{\delta_n \cdot g_n \quad w.p\ \alpha_n\ .
$$

The distortion of $\tilde{\mathcal{E}}$ is

$$\mathbb{I}(f(X) + W; \mathcal{E}|\tilde{\mathcal{E}}, X) = \mathbf{h}(f(X) + W|\tilde{\mathcal{E}}, X) - \mathbf{h}(f(X) + W|\mathcal{E}, X)$$

$$= \mathbf{h}\left(f(X) - \frac{\sqrt{c}}{r}\sum_{i=1}^{r} h_i(X) + W \Big| \tilde{\mathcal{E}}, X\right) - \mathbf{h}(W)$$

$$\overset{(a)}{\leq} \mathbf{h}\left(f(X) - \frac{\sqrt{c}}{r}\sum_{i=1}^{r} h_i(X) + W\right) - \mathbf{h}(W)$$

$$\overset{(b)}{\leq} \frac{M}{2}\ln\left(2\pi e\left(\sigma^2 + \mathbb{E}\left[\frac{\left\|f(X) - \frac{\sqrt{c}}{r}\sum_{i=1}^{r} h_i(X)\right\|^2}{M}\right]\right)\right) - \mathbf{h}(W)$$

$$= \frac{M}{2}\ln\left(1 + \frac{\mathbb{E}\left[\left\|f(X) - \frac{\sqrt{c}}{r}\sum_{i=1}^{r} h_i(X)\right\|^2\right]}{\sigma^2 M}\right)$$

$$\overset{(c)}{\leq} \frac{\mathbb{E}\left[\left\|f(X) - \frac{\sqrt{c}}{r}\sum_{i=1}^{r} h_i(X)\right\|^2\right]}{\sigma^2}$$

$$= \frac{\mathbb{E}\left[\left(\frac{\sqrt{c}}{r}\sum_{i=1}^{r} h_i(X)\right)^\top\left(\frac{\sqrt{c}}{r}\sum_{i=1}^{r} h_i(X)\right) - f(X)^\top f(X)\right]}{2\sigma^2}$$

$$= \frac{\mathbb{E}\left[\frac{c}{r^2}\sum_{i=1}^{r} h_i(X)^\top h_i(X) + \frac{c}{r^2}\sum_{i\neq j} h_i(X)^\top h_j(X) - f(X)^\top f(X)\right]}{2\sigma^2}$$

$$= \frac{\mathbb{E}\left[\frac{c}{r}\sum_{n=1}^{N} \alpha_i g_n(X)^\top g_n(X) + \frac{c(r-1)}{r} f(X)^\top f(X) - f(X)^\top f(X)\right]}{2\sigma^2}$$

$$\leq \frac{\mathbb{E}\left[c - f(X)^\top f(X)\right]}{2r\sigma^2}$$

$$\leq \frac{c}{2r\sigma^2}$$

$$= \epsilon,$$

where $(a)$ follows from conditioning reducing differential entropy, $(b)$ follows from Lemma 16, and $(c)$ follows from the fact that for all $m, x \in \Re_{++}, m\ln\left(1 + \frac{x}{m}\right) \leq x$.

We now upper bound the rate of $\tilde{\mathcal{E}}$.

$$\mathbb{I}(\alpha; \tilde{\mathcal{E}}) \leq \mathbb{H}(\tilde{\mathcal{E}})$$
$$\leq r \cdot \mathbb{H}(h_i)$$
$$\leq r\ln(2N)$$
$$= \frac{c\ln(2N)}{2\sigma^2\epsilon}$$

The result follows. $\qquad\square$

# E  Proof of Multilayer Results

**Lemma 19. (more is learned with the true input)** *Let $\tilde{\mathcal{E}}_{K:1}$ be a multilayer proxy. Then,*

$$\mathbb{I}(Y; \mathcal{E}_k|\mathcal{E}_{K:k+1}, \tilde{\mathcal{E}}_{k:1}, X) \leq \mathbb{I}(Y; \mathcal{E}_k|\mathcal{E}_{K:k+1}, \tilde{\mathcal{E}}_k, U_{k-1}).$$

*Proof.*

$$\mathbb{I}(\mathcal{E}_k; Y | \mathcal{E}_{K:k+1}, \tilde{\mathcal{E}}_{k:1}, X)$$

$$= \mathbb{I}(\mathcal{E}_k; \tilde{\mathcal{E}}_{k-1:1}, X, Y | \mathcal{E}_{K:k+1}, \tilde{\mathcal{E}}_k) - \mathbb{I}(\mathcal{E}_k; \tilde{\mathcal{E}}_{k-1:1}, X | \mathcal{E}_{K:k+1}, \tilde{\mathcal{E}}_k)$$

$$= \mathbb{I}(\mathcal{E}_k; \tilde{\mathcal{E}}_{j-1:1}, X, Y | \mathcal{E}_{K:k+1}, \tilde{\mathcal{E}}_k)$$

$$= \mathbb{I}(\mathcal{E}_k; Y | \mathcal{E}_{K:k+1}, \tilde{\mathcal{E}}_k) + \mathbb{I}(\mathcal{E}_k; \tilde{\mathcal{E}}_{k-1:1}, X | \mathcal{E}_{K:k+1}, \tilde{\mathcal{E}}_k, Y)$$

$$\overset{(a)}{\leq} \mathbb{I}(\mathcal{E}_k; Y | \mathcal{E}_{K:k+1}, \tilde{\mathcal{E}}_k) + \mathbb{I}(\mathcal{E}_k; \mathcal{E}_{k-1:1}, X | \mathcal{E}_{K:k+1}, \tilde{\mathcal{E}}_k, Y)$$

$$= \mathbb{I}(\mathcal{E}_k; \mathcal{E}_{k-1,1}, X, Y | \mathcal{E}_{K:k+1}, \tilde{\mathcal{E}}_k)$$

$$\overset{(b)}{=} \mathbb{I}(\mathcal{E}_k; Y | \mathcal{E}_{K:k+1}, \tilde{\mathcal{E}}_k, \mathcal{E}_{k-1:1}, X)$$

$$\overset{(c)}{=} \mathbb{I}(\mathcal{E}_k; Y | \mathcal{E}_{K:k+1}, \tilde{\mathcal{E}}_k, U_{k-1}),$$

where $(a)$ follows from the fact that $\mathcal{E}_k \perp \tilde{\mathcal{E}}_{k-1:1} | (X, Y, \mathcal{E}_{k-1:1})$ and the data processing inequality, $(b)$ follows from the fact that $\mathbb{I}(\mathcal{E}_k; \mathcal{E}_{k-1:1}, X | \mathcal{E}_{K:k+1}, \tilde{\mathcal{E}}_k) = 0$, and $(c)$ follows from the fact that $Y \perp (\mathcal{E}_{k-1:1}, X) | U_{k-1}$. □

Lemma 19 states that we learn more information about $\mathcal{E}_k$ when we are given the true input $U_{k-1}$ than when we are given $(X, \tilde{\mathcal{E}}_{k-1:1})$ and have to infer $U_{k-1}$. This is intuitive as we should be able to recover more about $\mathcal{E}_k$ when we observe its input exactly.

To prove 2), that the immediate output $U_k$ provides more information about $\mathcal{E}_k$ than does $Y$, each prototypical environment will involve slightly different analysis techniques. In essence, we expect a result like 2) to hold as a result of the data-processing inequality. No amount of post-processing of $U_k$ into $(Y)$ can increase the amount of available information conveyed about $\mathcal{E}_k$.

**Lemma 20.** *For all $K \in \mathbb{Z}_{++}, (N_0, N_1, \ldots, N_K) \in \mathbb{Z}_{++}^{K+1}$, and $\sigma^2 \geq 0$, if $N_K = 1$, and multilayer environment $\mathcal{E}_{K:1}$ consists of single-layer environments $\mathcal{E}_k$ that each identify a random function $f^{(k)}$ from $\Re^{N_{k-1}} \mapsto \Re^{N_k}$ for which*

$$L^{(k)} = \sup_{x, y \in \Re^{N_{k-1}}} \mathbb{E}\left[ \frac{\|f^{(k)}(x) - f^{(k)}(y)\|_2^2}{\|x - y\|_2^2} \Big| x = x, y = y \right] \leq 1,$$

*and*

$$U_k = \begin{cases} X & k = 0 \\ f^{(k)}(U_{k-1}) & 1 \leq k \leq K \end{cases}$$

*with $Y \sim \mathcal{N}(U_K, \sigma^2)$, then for any proxy $\tilde{\mathcal{E}}_k$,*

$$\mathbb{I}(Y; \mathcal{E}_k | \tilde{\mathcal{E}}_k, \mathcal{E}_{K:k+1}, U_{k-1}) \leq \frac{1}{2} \ln \left( 1 + \frac{\mathbb{E}\left[ \|U_k - \mathbb{E}[U_k | \tilde{\mathcal{E}}_k, U_{k-1}]\|_2^2 \right]}{\sigma^2} \right)$$

*Proof.*

$$\mathbb{I}(Y;\mathcal{E}_k|\tilde{\mathcal{E}}_k,\mathcal{E}_{K:k+1},U_{k-1}) = \mathbf{h}(Y|\tilde{\mathcal{E}}_k,\mathcal{E}_{K:k+1},U_{k-1}) - \mathbf{h}(Y|\mathcal{E}_K : k, U_{k-1})$$

$$= \mathbf{h}(Y|\tilde{\mathcal{E}}_k,\mathcal{E}_{K:k+1},U_{k-1}) - \mathbf{h}(W)$$

$$= \mathbf{h}\left(Y - (f_K \circ \ldots \circ f_{k+1})(\mathbb{E}[U_k|\tilde{\mathcal{E}},U_{k-1}])|\tilde{\mathcal{E}}_k,\mathcal{E}_{K:k+1},U_{k-1}\right) - \mathbf{h}(W)$$

$$\leq \mathbf{h}\left(Y - (f_K \circ \ldots \circ f_{k+1})(\mathbb{E}[U_k|\tilde{\mathcal{E}},U_{k-1}])\right) - \mathbf{h}(W)$$

$$\leq \frac{1}{2}\ln\left(1 + \frac{\mathbb{V}\left[U_K - (f_K \circ \ldots \circ f_{k+1})(\mathbb{E}[U_k|\tilde{\mathcal{E}},U_{k-1}])\right]}{\sigma^2}\right)$$

$$\leq \frac{1}{2}\ln\left(1 + \frac{\mathbb{E}\left[(U_K - (f_K \circ \ldots \circ f_{k+1})(\mathbb{E}[U_k|\tilde{\mathcal{E}},U_{k-1}]))^2\right]}{\sigma^2}\right)$$

$$\leq \frac{1}{2}\ln\left(1 + \frac{\mathbb{E}\left[\prod_{i=k+1}^{K} L^{(i)}\left\|U_k - \mathbb{E}[U_k|\tilde{\mathcal{E}}_k,U_{k-1}]\right\|^2\right]}{\sigma^2}\right)$$

$$\leq \frac{1}{2}\ln\left(1 + \frac{\mathbb{E}\left[\left\|U_k - \mathbb{E}[U_k|\tilde{\mathcal{E}}_k,U_{k-1}]\right\|^2\right]}{\sigma^2}\right)$$

$\square$

We now provide a proof of Theorem 10.

**Theorem 10. multilayer rate-distortion bound** *For all $K \in \mathbb{Z}_{++}$, $\sigma^2, \epsilon \geq 0$, if $\mathcal{E}_{K:1}$ is a multi-layer environment such that for all $k \in \{1, \ldots, K\}$ and $\delta \geq 0$, there exist $\tilde{\mathcal{E}}_k$ s.t.*

$$\mathbb{I}(Y;\mathcal{E}_k|\mathcal{E}_{K:k+1},\tilde{\mathcal{E}}_{k:1},X) \leq \mathbb{I}(U_k + W;\mathcal{E}_k|\tilde{\mathcal{E}}_k,U_{k-1}) \leq \delta,$$

*where $W \sim \mathcal{N}(0,\sigma^2 I)$, then*

$$\mathbb{H}_\epsilon(\mathcal{E}_{K:1}) \leq \sum_{k=1}^{K} \mathbb{H}_{\frac{\epsilon}{K}}(\mathcal{E}_k).$$

*Proof.* Let

$$\tilde{\Theta}_\epsilon^{K:1} = \{\tilde{\mathcal{E}} \in \tilde{\Theta}_\epsilon : \tilde{\mathcal{E}} = (\tilde{\mathcal{E}}_1, \ldots, \tilde{\mathcal{E}}_K); \tilde{\mathcal{E}}_i \perp \tilde{\mathcal{E}}_j \wedge \tilde{\mathcal{E}}_i \perp \mathcal{E}_j \text{ for } i \neq j\},$$

We have that

$$
\inf_{\tilde{\mathcal{E}} \in \tilde{\Theta}_\epsilon} \mathbb{I}(\mathcal{E}; \tilde{\mathcal{E}}) = \inf_{\tilde{\mathcal{E}} \in \tilde{\Theta}_\epsilon} \sum_{k=1}^K \mathbb{I}(\mathcal{E}_k; \tilde{\mathcal{E}} | \mathcal{E}_{K:k+1})
$$

$$
\overset{(a)}{\leq} \inf_{\tilde{\mathcal{E}} \in \tilde{\Theta}_\epsilon^{K:1}} \sum_{k=1}^K \mathbb{I}(\mathcal{E}_k; \tilde{\mathcal{E}} | \mathcal{E}_{K:k+1})
$$

$$
\overset{(b)}{=} \inf_{\tilde{\mathcal{E}} \in \tilde{\Theta}_\epsilon^{K:1}} \sum_{k=1}^K \mathbb{I}(\mathcal{E}_k; \tilde{\mathcal{E}}_k)
$$

$$
\overset{(c)}{\leq} \sum_{k=1}^K \inf_{\tilde{\mathcal{E}}_k \in \tilde{\Theta}_{\frac{\epsilon}{K}}^{(k)}} \mathbb{I}(\mathcal{E}_k, \tilde{\mathcal{E}}_k)
$$

$$
= \sum_{k=1}^K \mathbb{H}_{\frac{\epsilon}{K}}(\mathcal{E}_k)
$$

where $(a)$ follows from the fact that $\tilde{\Theta}_\epsilon^{K:1} \subset \tilde{\Theta}_\epsilon$, $(b)$ follows from the fact that $\tilde{\mathcal{E}}_i \perp \mathcal{E}_j$ for $i \neq j$, and $(c)$ follows from the fact that for $\tilde{\Theta}_{\epsilon/K}^{(k)} := \{\tilde{\mathcal{E}}_k \in \tilde{\Theta} : \mathbb{I}(U_k + W; \mathcal{E}_k | \tilde{\mathcal{E}}_k, U_{k-1}) \leq \frac{\epsilon}{K}\}$, $\tilde{\Theta}_{\epsilon/K}^{(1)} \times \ldots \times \tilde{\Theta}_{\epsilon/K}^{(K)} \subset \tilde{\Theta}_\epsilon^{K:1}$ since we assumed that $\mathbb{I}(Y; \mathcal{E}_k | \mathcal{E}_{K:k+1}, \tilde{\mathcal{E}}_{k:1}, X) \leq \mathbb{I}(U_k + W; \mathcal{E}_k | \tilde{\mathcal{E}}_k, U_{k-1})$ for all $k$. $\qquad \square$

## F   Proof of Main Results

**Theorem 11. (relu neural network rate-distortion and sample complexity bound)** *For all* $d, N, K \in \mathbb{Z}_{++}$ *and* $\sigma^2, \epsilon \geq 0$, *if multilayer environment* $\mathcal{E}_{K:1}$ *is the deep ReLU network with input* $X : \Omega \mapsto \Re^d$ *s.t.* $\mathbb{V}[X] = I_d$ *and output* $Y \sim \mathcal{N}(U_K, \sigma^2)$, *then*

$$
\mathbb{H}_\epsilon(\mathcal{E}_{K:1}) = \tilde{\mathcal{O}}\left(KN^2 + dN\right), \quad T_\epsilon = \tilde{\mathcal{O}}\left(\frac{KN^2 + dN}{\epsilon}\right).
$$

*Proof.* If we have a distortion function $d(\mathcal{E}_k, \tilde{\mathcal{E}}_k)$ for which

$$
\mathbb{I}(Y; \mathcal{E}_k | \mathcal{E}_{K:k+1}, \tilde{\mathcal{E}}_{k:1}, X) \leq d(\mathcal{E}_k, \tilde{\mathcal{E}}_k),
$$

then by the same proof as in Theorem 10,

$$
\mathbb{H}_\epsilon(\mathcal{E}_{K:1}) \leq \sum_{k=1}^K \mathbb{H}_{\frac{\epsilon}{K}}(\mathcal{E}_k, d),
$$

where $\mathbb{H}_{\frac{\epsilon}{K}}(\mathcal{E}_k, d)$ denotes the rate-distortion function for random variable $\mathcal{E}_k$ under distortion function $d(\mathcal{E}_k, \tilde{\mathcal{E}}_k)$. By Lemma 20, we have the above condition for $d(\mathcal{E}_k, \tilde{\mathcal{E}}_k) = \frac{1}{2} \ln\left(1 + \frac{\mathbb{E}[\|U_k - \mathbb{E}[U_k | \tilde{\mathcal{E}}_k, U_{k-1}]\|_2^2]}{\sigma^2}\right)$. Furthermore, distortion used in the proof of Theorem 17 is exactly $d(\mathcal{E}_k; \tilde{\mathcal{E}}_k)$. As a result,

$$
\mathbb{H}_\epsilon(\mathcal{E}_{K:1}) \leq \sum_{k=1}^K \mathbb{H}_{\frac{\epsilon}{K}}(\mathcal{E}_k, d)
$$

$$
\overset{(a)}{=} \mathcal{O}\left((K-2)N^2 \ln\left(\frac{NK}{2\sigma^2\epsilon}\right) + dN \ln\left(\frac{dK}{2\sigma^2\epsilon}\right) + N \ln\left(\frac{NK}{2\sigma^2\epsilon}\right)\right)
$$

$$
= \tilde{O}(KN^2 + dN).
$$

where $(a)$ follows from Theorem 17. The sample complexity result follows from applying Theorem 9. $\qquad \square$