# OpenReview forum: "An Information-Theoretic Framework for Deep Learning"
_NeurIPS.cc/2022/Conference — NeurIPS 2022 Accept_

### Official Review · Reviewer_rFcg · 2022-07-10

**Rating:** 4
**Confidence:** 5
**Soundness:** 2 fair
**Presentation:** 2 fair
**Contribution:** 2 fair

**Summary:**

The paper addresses a problem in Machine Learning very much deserving attention: to develop an information theoretic basis for Deep Learning (DL) Networks. Still largely heuristic, DL  would greatly benefit from an information theoretic perspective to help understand the complexity variation (or potentially required) of  a network with   breadth and depth. It promises in particular, an ability to consider several layers (i.e. varying depth) without necessarily invoking  an exponential dependence on the depth, with potentially infinite breadth with a constrained boundedness on the sum of the parameter weights.
Overall the paper is clearly written with a good description of its objectives and an acceptable review of the current lay of the land.


**Questions:**

1. Given that I was not privvy to the details of the proofs (no appendices), how does the independence assumption in IID property weaken the result?
2. For a given distribution, and if one could associate an energy functional to the data (as in the Gibbs distribution expression), could one guide the analysis in estimating the (even a bound) depth complexity?


**Ethics Review Area:**

["I don’t know"]

**Limitations:**

I have not really seen any development/statement of any significance in the societal impact direction.
They did mention that their paper had not addressed any practical issue related to their development.

 In addition, this reviewer did not think that the title of the paper was a adequately matched reflection of the results in the paper.

**Strengths And Weaknesses:**

The goal of the paper is very legitimate and hence worthy, and understanding the complexity of a network (sample and parameterization) would be in the same vein as model order estimation/complexity in the linear world. This is a problem, as duly noted in the paper, investigated by a number of researchers and each with their set of assumptions.
The objectives are nicely motivated and clearly stated.

Weaknesses:
While a good effort is expended at maintaining some rigor, which is very welcome in this area, given the plethora of data-driven and data-dictated efforts, I would sadly add that some sloppy notation and some mathematically erroneous/misleading  statements were strewn here and there. While I could not verify the proofs to the theorems, since I had no access to the Appendices, I found some of the key assumptions that i inferred to be key to the claimed strength of the results were also impractical and hence of limited interest (e.g. IID property of data, which is certainly not the case in practice).

---

> ### Author Response · Authors · 2022-07-31
> **Response to Reviewer rFcg**
>
> The reviewer comments on our “sloppy notation”. We apologize as even we know that our notation is unconventional to the machine learning audience. It is ultimately subjective as the other reviewers did not comment on the notation and reviewer ePAL even seemed to find it intuitive and readable.  We needed to develop notation that allowed for coherent use of both probability theory and information theory.  We are of course open to make changes if you identify any confusing abuses of notation.
>
> The reviewer also suggests that some mathematical statements are “erroneous”. We are not sure which results you are referring to. The appendix presents all proofs and is in the supplementary pdf file. We don’t know why you are not able to see it.
>
> Lastly, the reviewer addresses concerns about the iid assumption of data in our framework. We assume you are concerned with the iid assumption on X. We would firstly argue that this is a standard assumption in supervised learning, which is the scope of this paper. If it were not the case however, the independence assumption only impacts 1) the lower bounds on rate-distortion/sample complexity and 2) the monotonicity of error result. We think that in the context of this paper, these results are rather ancillary. We should explicitly make these statements clear in the paper though since it could be a concern to future readers.
>
> We don’t study nonstationary at all in this work so we don’t have much to say about the “identically distributed” component of “iid”. We suppose that in settings such as reinforcement learning we will not have iid X, but assuming iid episodes is a common workaround.
>
> We will note that we also assume that the *data pairs* are iid conditioned on the environment. We really just need that $\mathbb{I}(Y_{t+1};(X_s, Y_{s+1})|X_t, \mathcal{E}) = 0$ for $s \neq t$. This ought to be the case (why would additional data $(X_s, Y_{s+1})$ give you information about a label $(Y_{t+1})$ when you already know the input $(X_t)$ and the true labeling function $(\mathcal{E})$).
>
> We thank the reviewer for taking time to formulate challenges to our work. The process of thinking through which results rely on the iid assumption was very enlightening even for us the authors. We hope that the reviewer understands that such an assumption is in general not very restrictive in the supervised learning framework and particularly the independence assumption doesn’t have large implications for the results of our paper. With this in mind, we hope that the reviewer can recontextualize the results and contributions of our work.

---

> > ### Comment · Reviewer_rFcg · 2022-08-06
> > **Assumptions_ARE_OK**
> >
> > Making assumptions to overcome a possibly intractable problem is indeed quite reasonable and acceptable, it is, however, important to clearly state it as such, because the real world is not that. It helps enlighten a reader, particularly young theorists who are just getting into the field... the fact that many papers have done that, does not make it right. Pointing out what.where and how such a shortcoming might impact the current results would be quite acceptable.
> > I, personnally like this type of work.

---

> > > ### Author Response · Authors · 2022-08-06
> > > **Response to Reviewer rFcg**
> > >
> > > We agree wholeheartedly with the reviewer that such assumptions and their implications on results should be clearly stated for accessibility.
> > > Part of the value of the peer review process is comments like this which challenge the authors to stress-test their ideas and presentation against the concerns and confusions of readers. We'll be sure to make these points clear in future iterations of this work.

---

### Official Review · Reviewer_ePAL · 2022-07-10

**Rating:** 7
**Confidence:** 2
**Soundness:** 3 good
**Presentation:** 2 fair
**Contribution:** 3 good

**Summary:**

The authors proposed a novel information-theoretic framework with newly introduced notions of regret and sample complexity. The work is motivated by the gap between empirical success of generalizability of the deep neural networks and previous results on their complexity. By assuming an optimal agent and replace the out-of-sample error with average cumulative error, they provide general bounds on regret and sample complexity via rate-distortion theory. They further study the deep ReLU networks and deep non-parametric networks with the theory established earlier.

**Questions:**

The results in the paper are based on the "optimality" assumption, which may not be the case in the training process. This will change the monotonicity of the prediction errors. Can the authors elaborate on this?And does the current framework also provide a worth-case sample complexity?

**Limitations:**

The authors have included discussions on limitations and future directions in their paper.

**Strengths And Weaknesses:**

The authors focus on the average case rather than the worst case scenarios, and thus are able to show sample complexity grows linear or quadratic in network depth. This brings the theoretical understanding closer to what is observed in practice. The introduction of new notions are very intuitive and easy to understand.

The reference style needs to be fixed. Many of the references are not covered by parentheses properly (use \citep{} for example), so do equations (can use \eqref{}). Minor typos are observed, like line 77, "consider an our framework". And the presentation of the paper could be greatly improved if the authors add more cross references in the paper when mention some equations/notations they defined earlier in the paper.

---

> ### Author Response · Authors · 2022-07-31
> **Response to Reviewer ePAL**
>
> The reviewer asks about whether our monotonicity of error result would hold for a practical agent. Unfortunately, the scope of results in this paper is restricted to the theoretical limits of information that can be extracted from data. However, in the appendix (and alluded to in the text), there is Corollary 15 which quantifies the shortfall experienced by a suboptimal learning algorithm (i.e a tractable algorithm like sgd). The result provides an initial direction for future work to study the performance of practical algorithms.
>
> The reviewer also asks about whether our “monotonicity in expectation” results will also hold in a worst-case sense. By “worst” we are assuming you mean worst-case within a confidence set which covers $1-\delta$ probability. Classical Shannon information theory focuses on expectations, though one could easily derive loose high probability bounds via Markov’s Inequality. While these worst-case bounds would be nice to have, it’s becoming quite clear that such results are not predictive of the behavior of modern machine learning methods. We think of the average case framework as a feature rather than a drawback if it can lead to results that better mirror empirical results.
>
> We thank the reviewer for taking time to read our paper. Hopefully the above explanations adequately address the posed questions. We are pleased to see that the reviewer enjoyed and digested the main points of the paper and sees the value in the framework and results.

---

### Official Review · Reviewer_B2L8 · 2022-07-11

**Rating:** 5
**Confidence:** 3
**Soundness:** 2 fair
**Presentation:** 2 fair
**Contribution:** 3 good

**Summary:**

This paper proposes an information-theoretic framework that builds on rate-distortion theory for analyzing sample complexity of learning with an average-case notion of regret based on KL divergence. Under this framework, analysis and complexity bounds for two specific network setups are also provided.

**Questions:**

My questions are outlined above.

**Limitations:**

No.

**Strengths And Weaknesses:**

Strengths:
1. The main results motivated by the average-case analysis are interesting.
2. The analysis for two network environments is interesting.

Weaknesses:
1. Lack of empirical justification.
2. Organization of the paper could be improved.

Details:
1. The main weakness in my opinion is the omission of any experimental justification for the proposed framework. Particularly, I am really curious about how the average-case analysis that this framework builds on holds in reality. For example, to what extent does the quadratic depth dependence hold? Since the paper already provides relevant analysis for some prototypical networks, I think it would strengthen the paper a lot more if some preliminary results can be provided to demonstrate the relevance of this framework in the real world.
2. This paper builds on rate-distortion theory but related work provided in the paper is quite limited. A dedicated section of preliminaries would help improve the readability of the paper.

---

> ### Author Response · Authors · 2022-07-31
> **Response to Reviewer B2L8**
>
> The reviewer notes the lack of empirical justification in our work. Ideally we would have empirical results too, but there is no realistic way to include that within the space constraints. The paper already covers a large amount of ground introducing general regret and sample complexity bounds based on rate-distortion and delivers novel stronger bounds for ReLU and nonparametric networks as well as an arsenal of new tools to think about other problems in deep learning. The empirical results are certainly an exciting future research direction that we are currently actively pursuing in parallel (whether these theoretical results of an optimal learner apply for a learner that uses standard sgd with automatic hyperparameter tuning).
>
> The reviewer also comments on a lack of related work particularly regarding rate-distortion theory. We regret to skimp on related work since rate-distortion theory is not a widely studied topic in the machine learning community. We did our best to fit everything into the page limit and we provided the minimum amount necessary to produce a self-contained paper. Elements of Information Theory (Cover, Thomas) provides a comprehensive overview of the topic so in future edits we will at least point the reader to this text in the section where we introduce rate-distortion theory.
>
> We thank the reviewer for taking time to read our paper and voice concerns. Hopefully the reviewer can appreciate the novel substantial theoretical results of this paper despite the lack of empirical results and cursory coverage of rate-distortion theory which we can address in future versions of the paper (as described above).

---

> > ### Comment · Reviewer_B2L8 · 2022-08-09
> > **Reviewer Response**
> >
> > I thank the authors for the detailed responses. Although the lack of empirical justification is a pity, I find the average-case analysis very interesting. It is encouraging to know that the authors are actively experimenting with the framework. I have revised my rating to recommend for acceptance.

---

### Official Review · Reviewer_BqHU · 2022-07-12

**Rating:** 6
**Confidence:** 4
**Soundness:** 4 excellent
**Presentation:** 3 good
**Contribution:** 3 good

**Summary:**

This work proposes a novel information-theoretic framework for understanding deep learning. Under the proposed framework, the sample complexity of learning a deep neural network scales linearly with the number of layers rather than exponentially, a problem conventional PAC bounds suffer from.

The framework assumes target deep function can be treated as a sequence of “environments” — each environment transforms the output of the previous environment. Learning can be reduced to minimizing the mutual information between a sequence of proxy environments (i.e., the deep neural networks used to do the learning) and the true environment via samples from the true environment. The key insight is that, under some assumptions, the average-case error can be decomposed as a layer-wise sum of mutual information between the proxy environments and the true environments.

**Questions:**

See Strengths And Weaknesses

**Limitations:**

Most limitations are addressed.

**Strengths And Weaknesses:**

Using the average-case analysis is crucial for the reduction from exponential sample complexity to linear in depth. Introducing information-theoretical analysis into understanding function approximation feels like a promising direction and may lead to future works that further utilize tools from information theory to understanding deep learning.

**Problems**
- More discussion of the connection to information bottleneck would be nice to have because the characterization that it is just an “optimization” method is in my opinion inadequate. It is fundamentally a study of “representation” and it also provides sample complexity.
- Does not account for implicit regularization and inductive bias, both of which are key ingredients of deep learning, and it’s not clear how one could do it.
- Following the previous point and as pointed out by the authors themselves, the resulting bound is essentially a parameter counting bound which is better than VC by a factor of depth. This reduction on a high level comes directly from the fact of the average case nature of the analysis; however, I am not convinced that this reduction is what is needed to fully understand why deep learning works. Notably, this bound is likely still loose for all intents and purposes and probably would not correlate well with generalization error [1,2].
- Following the previous point, there is currently no empirical verification for the assumptions made. It would be good to see some experimental results that demonstrate the validity of the assumptions.
- Can you also take advantage of the fact that the input $X$ usually has low entropy? As it stands, the framework only makes assumptions about the ground truth labeling function.



**Typo**
-  244. “Leading to potentially .. information” missing an adjective?

**Reference**
1. Fantastic Generalization Measures and Where to Find Them. Jiang & Neyshabur et al.
2. In Search of Robust Measures of Generalization. Dziugaite & Drouin et al.

---

> ### Author Response · Authors · 2022-07-31
> **Response to Reviewer BqHU**
>
> The reviewer mentions that our results do not account for “implicit regularization  and inductive bias”. If I am not mistaken, this is a phenomenon observed/studied for neural networks trained with gradient methods. Our results concern an optimal agent with a correctly specified weight prior. There is no “implicit regularization” or “inductive bias” because there is “explicit regularization” enforced by the true prior. To study practical algorithms like sgd, a starting point is Corollary 15 of the appendix (also alluded to in the main text) where we quantify the shortfall incurred by a “suboptimal agent”.
>
> The reviewer also expresses concern that our results are likely loose. The bounds are certainly not loose for the problems that we are studying. If data is truly generated by a neural network with relu activations and iid weights, a bound that is linear in the parameter count is as tight as one could hope. However, it is a good point that data generating processes in the real world may be much simpler and hence require fewer samples to learn useful models. Perhaps they are neural networks with dependent weights. To this extent we study the nonparametric network which has dependent weights within the same layer, resulting in width-independence in the bound. We do not study weight-dependence across layers in this work however which could be interesting future work.
>
> The reviewer also suggests that empirical verification would strengthen the paper. While ideally we would like to include empirical results, there is no realistic way to include that within the space constraints. It is an exciting future research direction that we are currently actively pursuing in parallel (whether these theoretical results of an optimal learner apply for a learner that uses standard sgd with automatic hyperparameter tuning).
>
> Finally, the reviewer asks if we can take advantage of $X$ with low entropy in our framework. Firstly, this is an excellent observation regarding the entropy of the input distribution. This is a subtle point, but it is handled by the rate-distortion calculation. For instance, if we consider d-dimensional linear regression $Y = \theta^\top X$ but the input $X$ is only non-zero in the 1st dimension. In order to obtain low error (distortion), we will only need to acquire bits about the first component of theta. The resulting rate distortion and hence sample complexity will shrink by a factor of $d$. Not every situation will be as simple as this one, but the framework is amenable to producing the correct result if the user carefully characterizes  the rate-distortion function.
>
> We thank the reviewer for taking the time to critique our work. From the questions posed, it seems that the reviewer digested a large portion of the paper and its implications. The comments have helped us better understand potential points of confusion/subtleties that would be useful to explicitly highlight in a final version of this paper.

---

> > ### Comment · Reviewer_BqHU · 2022-08-04
> > **Thank you for the reply.**
> >
> > I thank the authors for the reply
> >
> > > To study practical algorithms like sgd, a starting point is Corollary 15 of the appendix (also alluded to in the main text) where we quantify the shortfall incurred by a “suboptimal agent”.
> >
> > Implicit biases are used to refer to both the regularization from SGD and the model's own inductive bias and how to characterize them is still largely an open problem. It is arguably one of the most important problems in deep learning. My understanding is that the proposed framework likely cannot make understanding it easier but that's ok (in a similar sense that we don't know how to specify a good prior for PAC-Bayes bounds). On the other hand, can you elaborate how one should interpret corollary 15 and how one could use it in analysis? I feel this could be a valuable addition to the paper.

---

> > > ### Author Response · Authors · 2022-08-06
> > > **Comment on Corollary 15**
> > >
> > > We thank the reviewer for the thoughtful response.
> > >
> > > Corollary 15 is restated below:
> > > For all $t \geq 0$ and $P_t = \pi(H_t, Z_t)$,
> > >
> > > $\mathbb{E}[d_{KL}(P_t^*\|P_t)|H_t] $
> > >
> > > $=\mathbb{E}[d_{KL}(P^*|\hat{P}_t)|H_t]$
> > >
> > > $ + \mathbb{E}[d_{KL}(\hat{P}_t|P_t)|H_t]$
> > >
> > > (please excuse the strange formatting, it seems that the open review latex parser is bugged).
> > > The LHS of the equals sign denotes the error of an algorithm $P_t$ which is allowed to depend on the data set $H_t$ and potentially some algorithmic randomness $Z_t$. The first term on the RHS denotes the error incurred by an optimal posterior predictive. All the results in this paper are for bounding this quantity. The second term on the RHS is the *shortfall incurred* by using an imperfect algorithm $P_t$ as opposed to the optimal posterior predictive. This shortfall turns out to be exactly the KL divergence between the posterior's prediction of the output and the algorithm's.
> > >
> > > A concrete way that one could try to use this for analysis is to first upper bound the shortfall term by the KL divergence between
> > > $\mathbb{P}(\theta|H_t)$ and $P_t(\theta)$, an algorithm that produces a posterior estimate of the weights $(\theta)$ from the observed data. This upper bound follows from the data processing inequality. A concrete algorithm to study could be Stochastic Gradient Langevin Dynamics (SGLD), which produces samples from this posterior distribution.

---

> > > > ### Comment · Reviewer_BqHU · 2022-08-09
> > > > **Thanks for the reply.**
> > > >
> > > > I see. I believe having further discussions of these topics in the appendix would be fruitful.
> > > >
> > > > In any case, my opinions have not been significantly changed. I believe this is a solid work with new insights. There are many questions left open and it remains to see how useful the framework can be, but I think it should be accepted.

---

### Meta-Review · Area_Chair_3kJ6 · 2022-08-26

**Recommendation:** Accept
**Confidence:** Less certain

**Metareview:**

While some reviewers have expressed some criticism for the possibility that some assumptions might be unrealistic, all the reviewers commented on the refreshingly novel approach that could lead to new directions of research. Hence, while not perfect, this is an exciting paper that should be accepted at the conference. Please take into account the reviewers' comments in preparing the camera-ready version, in particular the comments on the limitations of the proposed approach.

**Award:**

No

---

### Decision · Program_Chairs · 2022-09-14

Accept